

# Impacts of Active Satellite Sensors' Low-level Cloud Detection Limitations on Cloud Radiative Forcing

Yinghui Liu[1]

[1]Center for Satellite Applications and Research, NOAA/NESDIS, Madison, WI, USA

*Correspondence to*: Yinghui Liu (yinghui.liu@noaa.gov)

**Abstract.** Previous studies revealed that satellites sensors with the best detection capability identify 25-40% and 0-25% fewer clouds below 0.5 km and between 0.5–1.0 km, respectively, over the Arctic land. Quantifying the impacts of cloud detection limitations on the radiation flux are critical especially over the Arctic Ocean considering the dramatic changes in Arctic sea ice. In this study, the proxies of the space-based radar, CloudSat, and lidar, CALIPSO, cloud masks are derived based on

simulated radar reflectivity and cloud optical thickness using retrieved cloud properties from surface-based radar and lidar during the Surface Heat Budget of the Arctic Ocean (SHEBA) experiment. Limitations in low-level cloud detection by the space-based active sensors, and the impact of these limitations on the radiation fluxes at the surface and the top of atmosphere (TOA), are estimated. The results show that the combined CloudSat and CALIPSO product generally detects all clouds above 1 km, while detecting 25% (9%) fewer in absolute values below 600 m (600 m to 1 km) than surface observations. These

detection limitations lead to uncertainties in the monthly mean cloud radiative forcing (CRF), with maximum absolute monthly mean values of 2.7 Wm⁻² and 4.0 Wm⁻² at the surface and TOA, respectively. The uncertainties for individual cases are larger – up to 30 Wm⁻². Cloud information from only the CALIPSO or the CloudSat leads to larger cloud detection differences compared to the surface observations, and larger CRF uncertainties with absolute monthly means larger than 10.0 Wm⁻². These uncertainties need to be considered when radiation flux products from CloudSat and CALIPSO are used in climate and weather

studies.

## 1 Introduction

Clouds are an important modulator of the radiation flux at the surface and top of atmosphere (TOA). Major advances in understanding cloud processes in the climate system have been made, and the amount of uncertainty in cloud feedback has decreased by 50% in recent years as reported in the technical summary of the sixth assessment report of the Intergovernmental

Panel on Climate Change (Arias et al. 2021). However, cloud feedback still has the largest uncertainty among all climate feedback types (Arias et al. 2021). Climate prediction relies on a deepened understanding of clouds in the current climate system and how they evolve in the future. Arctic clouds are unique for being ubiquitous at low altitudes within and above a stable boundary layer, and for persistent mixed-phase stratiform clouds in the boundary layer (Shupe et al. 2006, Cesana et al. 2012). Cloud properties and cloud formation, maintenance and dissipation mechanisms in the Arctic stand out because of the



unique environment in the Arctic, e.g. extreme low temperatures, ubiquitous surface-based temperature and moisture inversions, and very limited local sources of cloud condensation nuclei and ice nuclei. The harsh environment in the Arctic makes in situ observations of Arctic clouds challenging. Coupled with the dramatic changes in the Arctic especially the Arctic sea ice in the last few decades, Arctic clouds may pose the largest challenges in improving our understanding cloud feedback

mechanisms (Vavrus et al. 2009, Tan et al. 2015, Arias et al. 2021).

Better understanding Arctic clouds requires accurate observations/measurements of three-dimensional cloud macrophysical and microphysical properties. Observations from surface-based instruments, and space-based passive and active sensors have been used to study Arctic cloud properties, as well as their climatology and inter-annual variabilities. These observations have their relative strengths and limitations. Surface observations with active lidar and radar have a superior capability to measure

cloud properties of the whole column (Shupe et al. 2011, Shupe 2011, Zhao and Wang 2010, Dong et al. 2010) and to resolve the diurnal cycle, but with poor spatial coverage. Cloud products from passive satellite sensors in the visible and infrared spectrum, e.g. AVHRR, MODIS and VIIRS, have good spatial coverage and long time series, which is critical for climate studies (Key et al. 2016). However, such data sets are limited in providing whole cloud vertical distributions due to signal saturation and have difficulties detecting all clouds in the polar regions especially at night (Liu et al. 2010). Observations from

space-based active radar and lidar potentially provide three-dimensional cloud properties (Stephens et al., 2002; Winker et al., 2009, Vaughan et al. 2009, Hunt et al. 2009); these products have been used to study global cloud spatial distributions and their temporal changes (Li et al. 2015, Naud et al. 2015, Devasthale et al. 2011, Huang et al. 2012, Mace et al. 2009, Mace and Zhang 2014, Sassen and Wang 2008, 2012, Liu et al. 2012a), along with longwave and shortwave radiation flux profiles and corresponding heating rates (L'Ecuyer et al. 2008, Henderson et al. 2013). In addition to their relatively limited spatial coverage

compared to passive satellite sensors, space-based radar, e.g. CloudSat, has issues with radar ground clutter while space-based lidar, e.g. the Cloud-Aerosol Lidar with Orthogonal Polarization (CALIOP) onboard the Cloud-Aerosol Lidar and Infrared Pathfinder Satellite Observations (CALIPSO), has signal attenuation issues from high-level clouds (Marchand et al. 2008, Winker et al. 2009, Blanchard et al. 2014, Liu et al. 2017, Christensen et al. 2013), both of which lead to detection limitations of clouds near the surface. Even the state-of-art combined satellite-based radar and lidar does not detect all low-level clouds

(Blanchard et al. 2014, Liu et al. 2017).

Over land in the Arctic, with collocated observations of cloud properties from surface-based radar and lidar as reference/truth and from satellite-based radar and lidar (CloudSat and CALIPSO), a few previous studies have shown that satellite-based radar and lidar detect slightly more clouds 2 km above mean sea level (a.m.s.l.), comparable cloud amounts between 1 km and 2 km, and fewer clouds below 1 km (Blanchard et al. 2014, Liu et al. 2017, Mioche et al. 2015, Huang et al. 2012, Protat et al.

2014). More specifically, the annual mean cloud fraction from space-based retrievals shows 25–40% fewer clouds below 0.5 km in absolute value than those from surface-based observations, and fewer clouds from 0.5 to 1 km (Liu et al. 2017). A similar conclusion is expected over the Arctic Ocean due to the same detection limitations of space-based radar and lidar over the snow/ice covered ocean as over land. However, this theory has not been confirmed because of the lack of collocated surface-based and space-based radar-lidar observations over the Arctic Ocean. These cloud detection limitations by the space-based



combined radar and lidar are expected to introduce uncertainties in the surface radiation flux, uncertainties which also have not yet been quantified partially because there have not been enough collocated radiation flux estimates. With the recent dramatic changes in Arctic sea ice (Serreze et al. 2015) and the essential role of clouds in modulating sea ice growth (Boucher et al. 2013, Kay and L'Ecuyer 2013, Taylor et al. 2015), it is desirable to study the accuracy/uncertainty of cloud detection

and consequent radiation flux uncertainties over the Arctic Ocean from combined space-based radar and lidar measurements. To address the lack of collocated surface-based and space-based combined radar-lidar observations over the Arctic Ocean, one approach is to collect large amounts of cloud observations from surface-based radar and lidar and retrievals of cloud properties, including cloud phase, effective radius and water content, from these observations. These cloud properties can be used as inputs in a radiative transfer model to simulate radar reflectivity and cloud optical thickness, and to derive the proxies of cloud

masks from space-based lidar and radar individually, as well as in combination. Comparisons of the derived cloud masks to surface cloud observations as truth can be used to assess the satellite sensors' cloud detection limitations especially near the surface over the Arctic Ocean. The retrieved cloud properties can also be used as inputs in another radiative transfer model to compute the radiation fluxes with all clouds and with only those clouds detected by space-based radar and lidar to estimate the uncertainties in radiation flux due to space-based radar/lidar cloud detection limitations.

During the Surface Heat Budget of the Arctic Ocean (SHEBA) experiment, a year's worth of surface-based radar and lidar observations of clouds were collected from October 1997 to October 1998 (Uttal et al. 2002), and vertical profiles of cloud properties were retrieved based on those observations (Shupe et al. 2011). The SHEBA experiment provides a product of 1-minute-interpolated vertical distributions of cloud phase, cloud effective radius, and cloud water content from the surface to 23 km. In our study, this dataset serves as inputs to QuickBeam, a multipurpose radar simulation package to simulate the

CloudSat reflectivity, after which the CloudSat cloud mask is derived, followed by an equation to calculate the cloud optical thickness for CALIPSO, and then the CALIPSO cloud mask. A cloud mask for combined CloudSat and CALIPSO is then derived. Through this process we generate a data set of collocated cloud observations over the Arctic Ocean from surface-based and satellite-based combined lidar and radar to confirm the space based lidar and radar cloud detection limitations near the surface, and to assess the uncertainties in radiation flux at the surface and TOA due to these limitations.

**2 Data and Method**

During the SHEBA experiment, the key instruments for the cloud observations were surface-based lidar and radar. The lidar system was the Depolarization and Backscatter Unattended Lidar (DABUL) at a green wavelength (0.523 μm); the radar was the 35 GHz millimeter cloud radar (MMCR). The radar/lidar combined approach has been used to study the cloud occurrence, cloud microphysical properties, and their radiative impact on the surface (Intrieri et al. 2002a, Intrieri et al. 2002b, Shupe and

Intrieri 2004). A multi-sensor, fixed-threshold cloud phase classification scheme (Shupe 2007) has been developed and applied to 1-min-interpolated observations from the DABUL, MMCR, Microwave Radiometer, and radiosondes to distinguish each cloud pixel/layer as one of the following 10 categories: clear, ice, snow, liquid, drizzle, liquid cloud+drizzle, rain, mixed phase,





haze, and uncertain, and to retrieve the cloud effective radius and cloud water content for ice clouds and liquid clouds, respectively. Complete details can be found in Shupe (2007). The temporal frequency of the retrieval products is 1 minute, and the vertical coverage is from 150 m to 22950 m, with every 63 m from 150 to 1050 m and every 100 m above 1050 m. In our study, this dataset served as both the true/reference cloud mask and cloud properties, the inputs of a radar simulation

package to simulate the CloudSat reflectivity, inputs to calculate the cloud optical thickness, and also inputs of another radiative transfer model to compute radiation flux. However, we only included the profiles of clear, ice, liquid, and mixed phase clouds, excluding any vertical profiles including snow, drizzle, liquid cloud+drizzle, rain, haze, or uncertain retrievals. Figure S1 shows the vertical profiles of cloud phases on November 21, 1997 during the SHEBA experiment. The retrieved cloud effective radius and cloud water content are shown in Figure S2.

The CloudSat cloud profiling radar (CPR) transmits a pulse at 94 GHz and measures the returning backscattered energy which contains information on its interactions with cloud and precipitation particles and atmospheric gases. Its relative higher frequency suggests great sensitivity to both cloud particles and water vapor besides precipitation. QuickBeam is a multipurpose radar simulation package that can be used to simulate the vertical radar reflectivity for CloudSat CPR and other sensors (Haynes et al. 2007). The inputs include a profile of hydrometeor mixing ratios, hydrometeor distribution type, hydrometeor phase and

density, radar frequency, radar location (space based or surface based), and a temperature and moisture profile. In this study, QuickBeam was used to simulate the space-based CloudSat vertical reflectivity at 94 GHz with the 1-min-interpolated cloud property vertical distributions from SHEBA. For the distribution of ice and liquid clouds, we used the modified gamma distribution with a distribution width of 2 (Marchand et al. 2009, Haynes 2007), and results were similar with lognormal distribution. The mixing ratios were set to use the cloud ice/liquid water content, and the cloud phase comes from the 1-min-

interpolated cloud phase. It should be noted again that only profiles of ice, liquid, and mixed phase clouds were used for the simulation and subsequent statistical analysis, and all other profiles containing snow, drizzle, liquid cloud+drizzle, rain, haze, or uncertain were not included. The simulated CloudSat reflectivity on November 21, 1997 is shown in Figure 1a.

The primary instrument onboard CALIPSO is a near-nadir view lidar, the CALIOP. CALIOP uses three receiver channels: one measures the 1064 nm backscatter intensity and two channels measure orthogonally polarized components of the 532 nm

backscattered signal. The CALIOP can penetrate, and thus detect, clouds with optical thicknesses up to 5 (Winker et al. 2009). The cloud optical thickness was calculated based on retrieved cloud effective radius and cloud water content. For ice clouds, the cloud optical thickness, $\tau_i$, was calculated by expressions of the form

$$\tau_i = \int_{z_1}^{z_2} IWC\, dz \times \left(a_i + \frac{b_i}{r_e}\right) \quad (1)$$
(1)

where $z_1$ and $z_2$ are cloud base and cloud top heights, IWC is the ice water content, $r_e$ is the cloud effective radius, $a_i$ and $b_i$ are

constants and i is the spectral interval. For CALIOP wavelength, $a_i$ and $b_i$ are $3.448 \times 10^{-3}$ $m^2 g^{-1}$ and $2.431$ $\mu m m^2 g^{-1}$ respectively. Details of this approach can be found in Ebert and Curry (1992). For liquid clouds, the cloud optical thickness was calculated following

$$\tau = \frac{3\int_{z_1}^{z_2} LWC}{2r_e \rho_w}$$
(2)



where LWC is the liquid water content and $\rho_w$ is the water density (Dong et al. 1998). The vertical distribution of the estimated cloud optical thickness on November 21, 1997 is shown in Figure 1b.

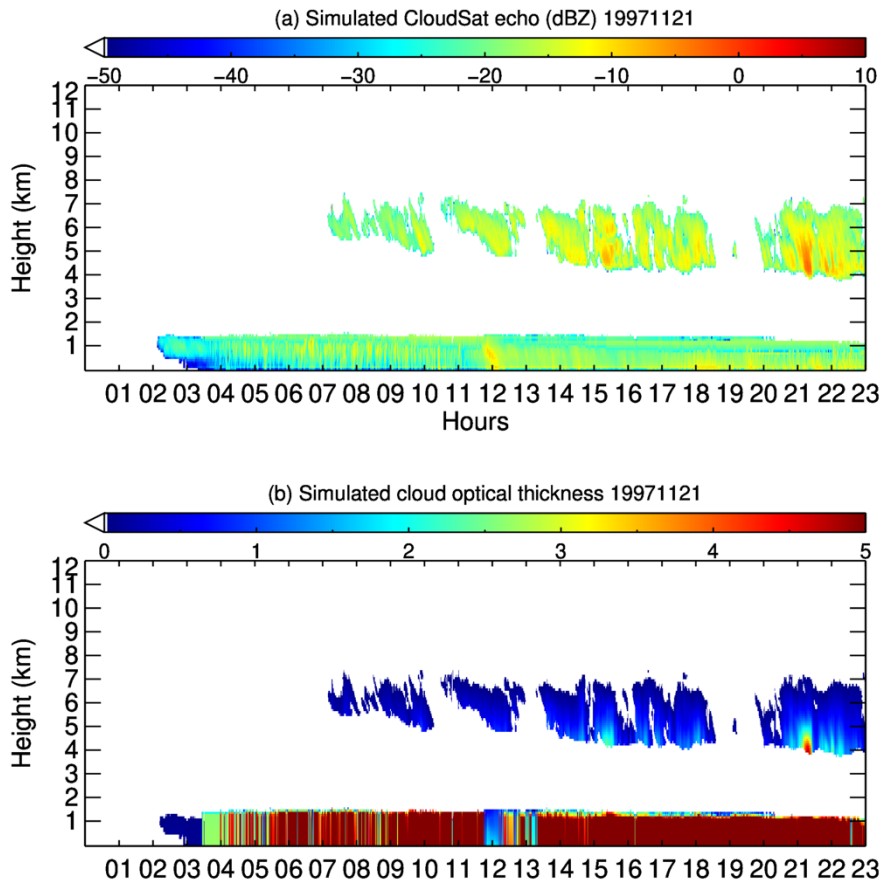

Figure 1: Simulated (a) CloudSat reflectivity and (b) integrated optical thickness from top of atmosphere for CALIPSO on
5    November 21, 1997 during the Surface Heat Budget of the Arctic Ocean (SHEBA) experiment.

With the simulated vertical profiles of CloudSat reflectivity and cloud optical thickness, cloud masks for CloudSat and CALIPSO were derived with the following approach. A layer was flagged as cloud if the simulated CloudSat reflectivity at a layer is larger than a set threshold, the mean radar-measured noise power. The mean radar-measured noise power can be
10    represented as the 99[th] percentile of the clear-sky returns as shown in Figure 7 of Marchand et al. (2008), Figure 2 in this paper. This threshold provides a very stringent requirement for cloud detection, especially for cloud detection near the surface. Since it was suggested that the CloudSat cloud detection capability would be improved with lower mean radar-measured noise power near the surface (Marchand et al. 2008), in this study, the mean radar-measured noise power was further lowered by 15 dBZe in the lowest 5 range bins (lower than 960 m) than the 99[th] percentile of the clear-sky returns while higher than or equal to -26



dBZe (Marchand et al. 2008). For the CALIPSO cloud mask, any layer with a calculated cloud optical thickness larger than 0 and accumulated cloud optical thickness from TOA less than 5 above this layer was flagged as cloud. In the cloud mask with the combined CloudSat and CALIPSO, a layer was flagged as cloud if the layer was flagged as cloud by either sensor. The cloud mask on November 21, 1997 from CALIPSO, CloudSat, and combined CALIPSO and CloudSat (CC) are shown in

5  Figure 3. In this study, we calculated and examined the vertical profiles of cloud fraction from CALIPSO, CloudSat, and combined CALIPSO and CloudSat at layers from 150 m to 12 km. The mean cloud (ice cloud, liquid cloud, and mixed phase cloud) fraction at a vertical layer was calculated as the ratio of the number of profiles identified as cloud (ice cloud, liquid cloud, and mixed phase cloud) at this layer compared to the total number of profiles.

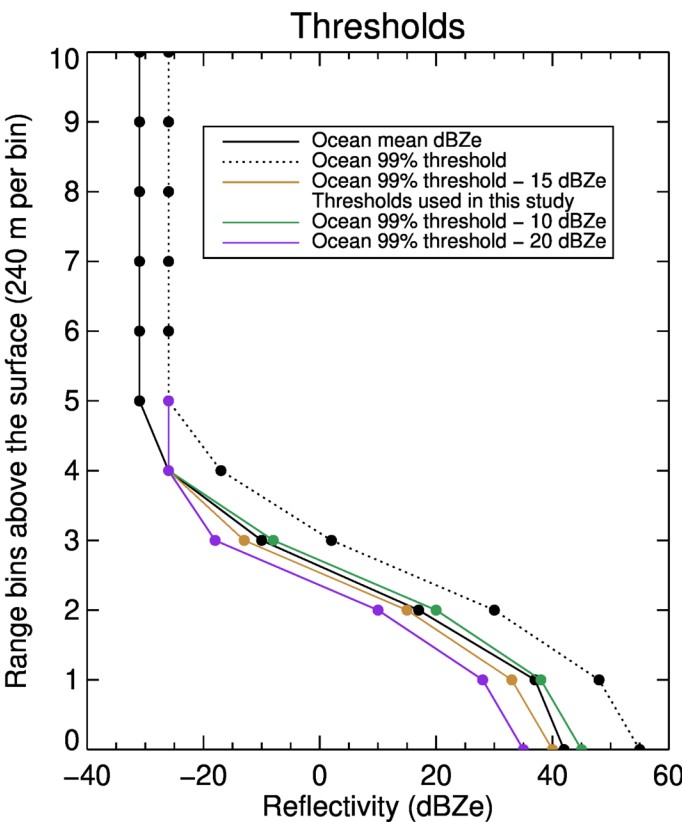

10  Figure 2: Typical (estimated) surface clutter profile adapted from Figure 7 in Marchand et al. (2008) and thresholds used (blue line) in this study to detect clouds for CloudSat.



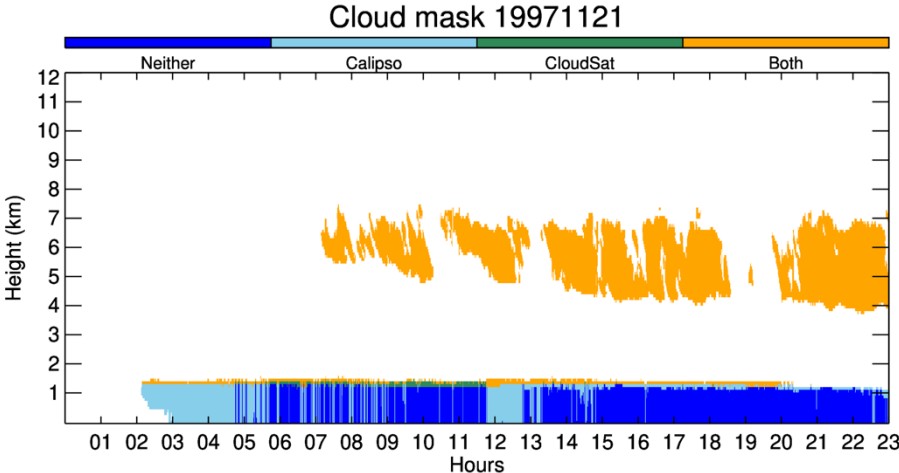

Figure 3: Cloud mask vertical profile based on simulated CloudSat reflectivity and cloud optical thickness for CALIPSO on November 21, 1997 collected during the Surface Heat Budget of the Arctic Ocean (SHEBA) experiment.

Two sets of cloud profiles were used as inputs to a radiative transfer model to calculate the radiation fluxes, including downward and upward longwave and shortwave radiation, shortwave and longwave cloud radiation fluxes, and cloud radiative forcing (CRF) at the surface and TOA. CRF is defined as the difference between the all-sky and clear-sky net radiation fluxes at the surface or TOA, which measures a cloud's impacts on the surface or TOA. One set was the complete cloud profiles from the surface observations, including cloud classification, cloud effective radius, and cloud water content. The other set was a

subset of the cloud profiles including only those layers identified as clouds in the space-based radar, lidar, or combined lidar and radar cloud mask. The radiative transfer model is Streamer (Key and Schweiger 1998). Streamer can compute both radiance and irradiance under a wide variety of atmospheric and surface conditions (Liu et al. 2004, Loyer et al. 2021). A discrete ordinate solver is used to calculate the longwave and shortwave radiation fluxes. The surface type is selected as fresh snow, and the albedo is from a built-in spectral albedo model scaled by the surface broadband albedo from the surface

observations. Albedo were computed for each hour during SHEBA by the Atmospheric Surface Flux Group (ASFG) radiometers (Intrieri et al. 2002a, Persson et al. 2002) (Figure 4). The monthly mean surface broadband albedos are lowest in July during the SHEBA experiment (Table 1, Figure 4). The albedo for the cloud profile was set as the closest ASFG albedo. Streamer can simulate the radiation fluxes for up to 50 cloud layers. In this study, only cloud layers from 150 m to 12.0 km are simulated. There are 125 layers from 150 m to 12.0 km in the retrieved cloud data sets, so that there are potential 125 cloud

layers at maximum. Every cloud layer below 2.0 km was included in the Streamer input file, with cloud top height, cloud physical thickness, cloud effective radius, and cloud water content; for cloud layers above 2.0 km, these cloud parameters were calculated for every 5 layers, including cloud top height, cloud thickness, mean cloud effective radius and water content for water and ice clouds, respectively. Hexagonal solid column was chosen as the shape of the ice clouds, and sphere was chosen as the shape of liquid clouds. Temperature and moisture profiles come from the closest radiosonde, which were launched at



least twice daily during the SHEBA experiment. The available cloud profiles have a temporal frequency of 1 minute. In this study, the radiation fluxes are computed using profiles with 1-hour intervals (one out of 60 profiles), and there are 24 cases in a day. Daily means are computed based on the 24 values, and the monthly means are calculated based on the daily means.

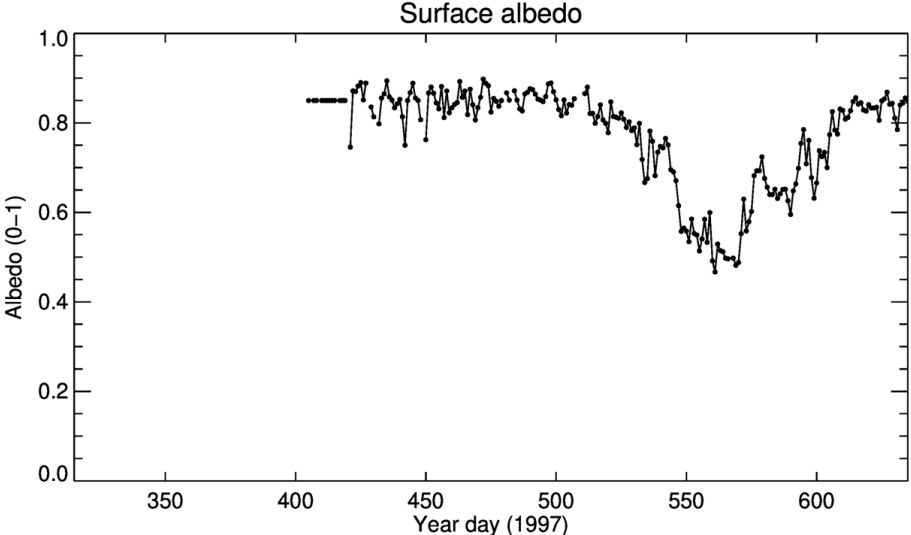

5    Figure 4: Annual cycle of the surface broadband albedo during the Surface Heat Budget of the Arctic Ocean (SHEBA) experiment.

- ▪ Table 1: Monthly mean surface broadband albedo during the Surface Heat Budget of the Arctic Ocean (SHEBA) experiment.

| Month | Oct | Nov | Dec | Jan | Feb | Mar | Apr | May | Jun | Jul | Aug | Sept |
|---|---|---|---|---|---|---|---|---|---|---|---|---|
| Year | 1997 | 1997 | 1997 | 1998 | 1998 | 1998 | 1998 | 1998 | 1998 | 1998 | 1998 | 1998 |
| Starting Julian Day | 274 | 305 | 335 | 366 | 397 | 425 | 456 | 486 | 517 | 547 | 578 | 609 |
| Ending Julian Day | 304 | 334 | 365 | 396 | 424 | 455 | 485 | 516 | 546 | 577 | 608 | 638 |
| Mean Albedo | 0.85 | N/A | N/A | N/A | 0.85 | 0.84 | 0.85 | 0.85 | 0.76 | 0.55 | 0.69 | 0.84 |



The uncertainties in the radiation fluxes due to the omission of clouds from the space-based radar, lidar, or combined radar and lidar were estimated as the difference between the radiation fluxes with partial and complete cloud profiles; all inputs of the radiative transfer model were the same for these two fluxes except inclusion/exclusion of the cloud layers that the space-based radar, lidar, or combined radar and lidar does not detect. The differences in these two fluxes can be used to quantify the impact of the space-based radar, lidar, or combined radar and lidar cloud detection limitations on the radiation fluxes at the surface and TOA. Rather than investigating all radiation fluxes, e.g. downward and upward longwave and shortwave radiation fluxes, this study focused on quantifying the uncertainties in the CRFs at the surface and TOA. In this paper hereinafter, the differences in the CRFs means the differences between the CRFs computed with the complete cloud profiles from the surface observations and those computed with the cloud profiles with layers identified as clouds by the CloudSat, CALIPSO, or the combined CloudSat and CALIPSO unless otherwise noted. The difference in the cloud amount hereinafter is the difference compared to the clouds (or cloud percentage) from the surface observations. The space-based radar, lidar, or combined radar and lidar are interchangeable with CloudSat, CALIPSO, or combined CloudSat and CALIPSO.

## 3 Results

### 3.1 Cloud fraction vertical distributions

Mean cloud vertical distributions from SHEBA's surface observations show higher cloud amounts closer to the surface, mainly below 1 km (Figure 5, same as Figure 3b in Shupe et al. 2011). The high values near the surface show two maxima, one in August and September around at 80%, and another in April and May around 70%. Mean cloud amounts above 2 km show high values in April, September, and in July towards higher altitudes. Local minima appear between 1.5 and 4 km in January, February, June and July, and October.

Above 1 km, CloudSat detects most of the clouds identified via surface observations (Figure 6b), with most of the differences between 0.0% to -5.0% (Figure 6e). The exceptions are in April, when CloudSat shows much lower cloud amounts between 1 km and 8 km, with the differences between -5.0% to -8.0%. These negative differences are collocated with the higher amounts of ice clouds (Figure 7a), consistent with the fact that CloudSat has limited sensitivity to optically thin ice clouds (Stephens et al. 2002). CALIPSO also detects most of the clouds seen from surface sensors (Figure 6a), with most of the differences between 0.0% and -5.0%, except that CALIPSO detects much fewer clouds between 1 km and 4.5 km from June to September, with maximum negative differences in September. These large negative differences are collocated with the higher amount of liquid and mixed phase clouds between 1 km and 6 km from June to September (Figure 7d, 7g), which have very high cloud optical thickness and saturate the CALIPSO signal for the clouds below. The combined CloudSat and CALIPSO detects all the clouds that surface observations show (Figure 6c, 6f), with differences between 0.0% and -1.0%, except for fewer clouds between 1 km and 2 km in August and September with differences between -8.0% and -3.0%.





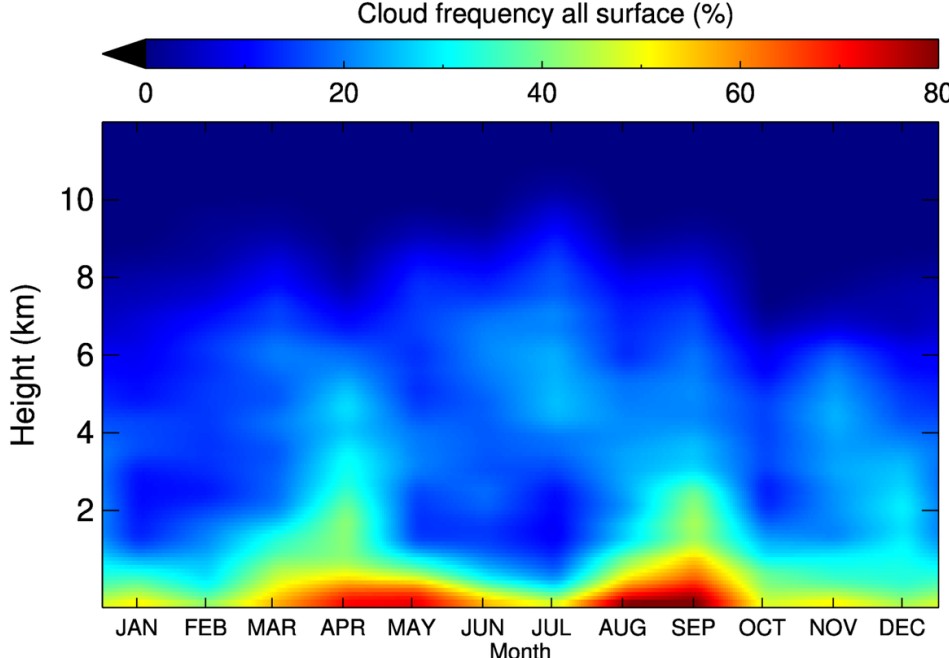

Figure 5: Mean cloud vertical distribution from the surface observations from October 1997 to October 1998 during the Surface Heat Budget of the Arctic Ocean (SHEBA) experiment.

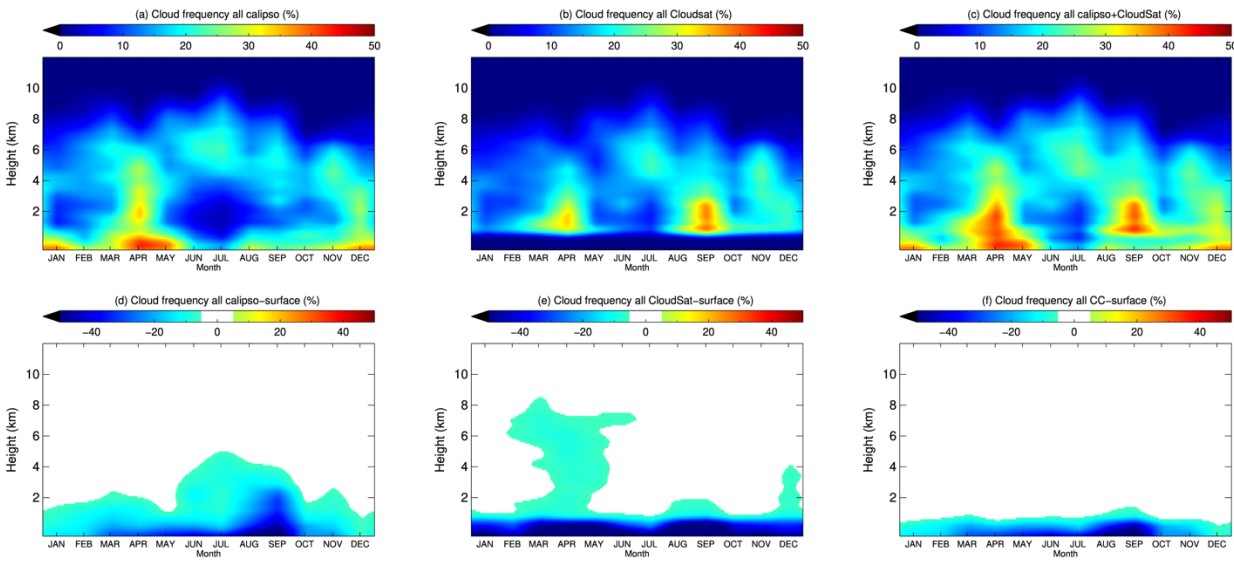

5    Figure 6: Cloud vertical distributions from (a) CALIPSO, (b) CloudSat, (c) combined CloudSat and CALIPSO, and the difference between (d) CALIPSO, (e) CloudSat, and (f) combined CloudSat and CALIPSO and surface observations during the Surface Heat Budget of the Arctic Ocean (SHEBA) experiment.





Below 1km, the CloudSat detects much fewer clouds than the surface observations because of surface clutter (Figure 7e). The percentage of clouds that CloudSat detects compared to the surface observations is 0% below 600 m for all months and gradually increases to around 75% near 1 km for most months, with similar results for ice, liquid, and mixed phase clouds for

all months. CALIPSO detects some clouds under 1 km depending on the accumulated cloud optical thickness above. In winter, the most common cloud types are ice clouds, which have relatively small cloud optical thicknesses; thus, CALIPSO can detect some clouds near the surface, e.g. around 80% in December and January and around 60% in other winter months. While in summer, the liquid and mixed phase clouds at higher levels often have high optical thicknesses and saturate the CALIPSO signal above 1 km, thus CALIPSO detects limited clouds near the surface, e.g. around 30% in September and 40% in other

summer months. The combined CloudSat and CALIPSO approach takes advantage of the detection capability of both CloudSat and CALIPSO. However, because of the poor detection capability of CloudSat near the surface, especially below 600 m, the combined CloudSat and CALIPSO detects similar amounts of clouds as that of CALIPSO alone, especially below 600 m.

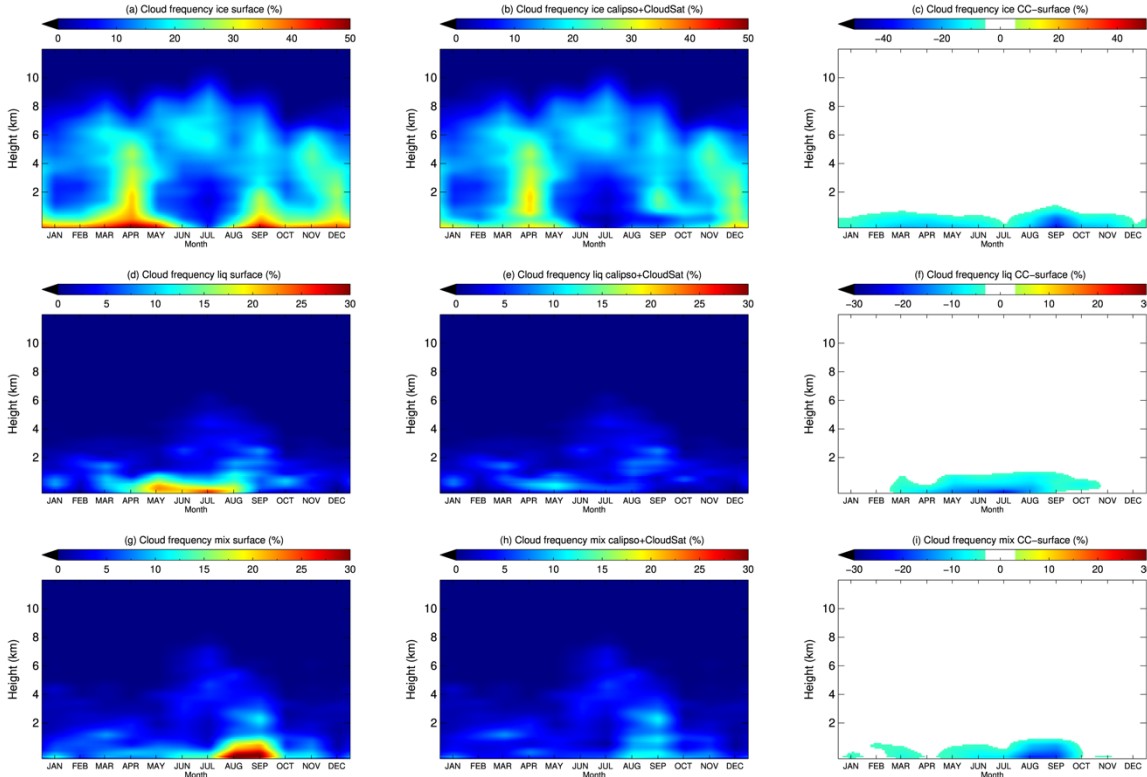

Figure 7: Cloud vertical distribution from surface observations for (a) ice (d) liquid (g) mixed phase cloud, from combined

CloudSat and CALIPSO for (b) ice (e) liquid (h) mixed phase cloud, and their differences for (c) ice (f) liquid (i) mixed phase cloud during the Surface Heat Budget of the Arctic Ocean (SHEBA) experiment.



The annual mean cloud amount vertical profiles (Figure 8, Table 2) show similar features for cloud detection capabilities from CloudSat, CALIPSO, and combined CloudSat and CALIPSO as those illuminated in the time-height cloud amount distributions (Figure 6, 7). Below 600 m, the CloudSat cloud amount is 0.0%, so the combined CloudSat and CALIPSO cloud detection capability comes from the CALIPSO, which detects roughly half of the clouds detected by the surface observations,
5 25% less in mean absolute values (Table 2). From 600 m to 1 km, CloudSat's detection capability increases, while CALIPSO sees slightly above 50% of all the clouds detected by surface observations, and the differences between the combined CloudSat and CALIPSO cloud amount and those from surface observations drop from -16.2% to -2.7%, roughly 9% less in mean absolute values. From 1 km to 2 km, the CALIPSO and CloudSat detection capabilities continue to improve, and their combination detects almost all the clouds seen in surface observations with differences between -2.7% and -1.3%. Above 2
10 km, the CloudSat has slightly lower cloud amounts than that from the surface due to thin ice clouds that remain undetected, while CALIPSO's detection capability increases because of the smaller accumulated cloud optical thickness above. As the result, the combined CloudSat and CALIPSO detects most of the clouds that surface observation sees above 2 km.

▪ *Table 2: Mean cloud amount from surface observations, CALIPSO, CloudSat, combined CALIPSO and CloudSat*
15 *(CC), cloud amount difference of CALIPSO and surface, CloudSat and surface, and combined CALIPSO and CloudSat (CC) and surface based on data from the Surface Heat Budget of the Arctic Ocean (SHEBA). Values are shown for all other layers between 149.5 m and 2050.0 m, every 5 layers between 2050.0 m and 4050.0 m, and every 10 layers between 4050.0 m and 12050.0 m. These values are the same as those shown in Figure 8.*

| Height (m) | Surface (%) | Calipso (%) | CloudSat (%) | CC (%) | Calipso – Surface | CloudSat - Surface | CC - Surface |
|---|---|---|---|---|---|---|---|
| 149.5 | 59.7 | 28.9 | 0.0 | 28.9 | -30.7 | -59.7 | -30.7 |
| 275.5 | 57.9 | 29.6 | 0.0 | 29.6 | -28.3 | -57.9 | -28.3 |
| 401.5 | 51.6 | 27.7 | 0.0 | 27.7 | -23.9 | -51.6 | -23.9 |
| 527.5 | 42.8 | 22.7 | 0.0 | 22.7 | -20.2 | -42.8 | -20.2 |
| 653.5 | 39.0 | 21.2 | 1.8 | 22.8 | -17.8 | -37.2 | -16.2 |
| 779.5 | 37.1 | 20.5 | 13.1 | 27.6 | -16.6 | -24.0 | -9.5 |
| 905.5 | 31.9 | 17.2 | 21.2 | 27.7 | -14.7 | -10.7 | -4.2 |
| 1050.0 | 30.4 | 16.9 | 23.0 | 27.6 | -13.4 | -7.4 | -2.7 |
| 1250.0 | 25.6 | 14.1 | 20.2 | 23.5 | -11.5 | -5.4 | -2.1 |
| 1450.0 | 24.2 | 13.5 | 19.4 | 22.4 | -10.7 | -4.8 | -1.8 |
| 1650.0 | 24.1 | 13.8 | 19.3 | 22.5 | -10.3 | -4.8 | -1.6 |
| 1850.0 | 23.3 | 13.8 | 18.8 | 22.0 | -9.5 | -4.5 | -1.3 |
| 2050.0 | 22.4 | 13.6 | 18.2 | 21.1 | -8.9 | -4.2 | -1.3 |




| | | | | | | | |
|---|---|---|---|---|---|---|---|
| 2550.0 | 21.4 | 14.2 | 18.1 | 20.8 | -7.2 | -3.3 | -0.7 |
| 3050.0 | 21.5 | 16.0 | 17.2 | 20.5 | -5.0 | -3.9 | -0.6 |
| 3550.0 | 20.8 | 17.0 | 16.9 | 20.5 | -3.8 | -4.0 | -0.4 |
| 4050.0 | 20.2 | 17.6 | 16.2 | 20.0 | -2.6 | -4.0 | -0.2 |
| 5050.0 | 18.9 | 17.8 | 15.1 | 18.9 | -1.1 | -3.8 | -0.0 |
| 6050.0 | 16.9 | 16.7 | 12.6 | 16.9 | -0.2 | -4.2 | -0.0 |
| 7050.0 | 12.5 | 12.5 | 8.3 | 12.5 | -0.0 | -4.3 | -0.0 |
| 8050.0 | 7.9 | 7.9 | 4.7 | 7.9 | -0.0 | -3.2 | 0.0 |
| 9050.0 | 3.2 | 3.2 | 1.5 | 3.2 | -0.0 | -1.7 | 0.0 |
| 10050.0 | 0.7 | 0.7 | 0.3 | 0.7 | 0.0 | -0.4 | 0.0 |
| 11050.0 | 0.1 | 0.1 | 0.0 | 0.1 | 0.0 | -0.0 | 0.0 |
| 12050.0 | 0.0 | 0.0 | 0.0 | 0.0 | 0.0 | -0.0 | 0.0 |

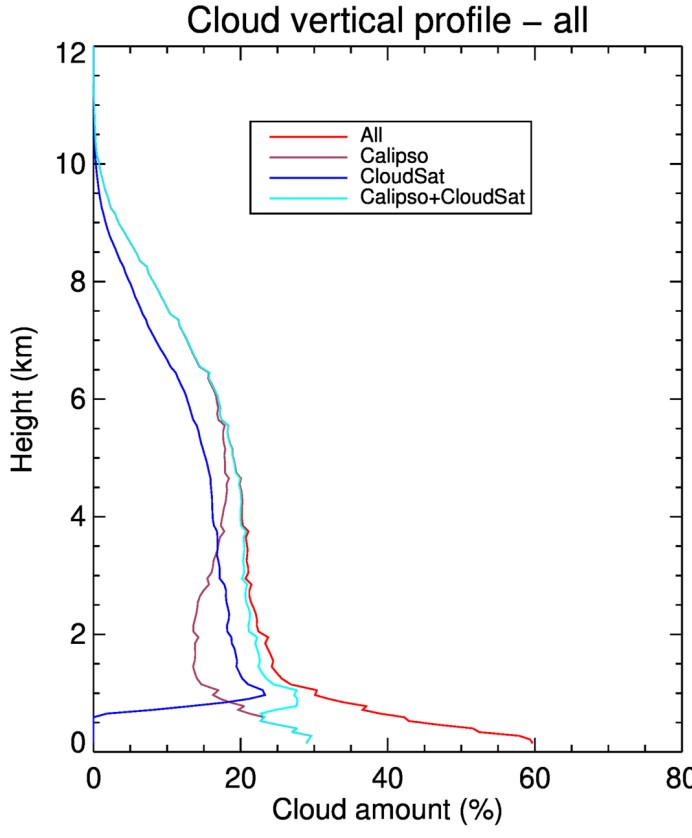



Figure 8: Mean cloud amount vertical distributions from surface observations, simulated CloudSat, CALIPSO, and combined CloudSat and CALIPSO during the Surface Heat Budget of the Arctic Ocean (SHEBA) experiment.

Ice clouds have a similar time-height distribution as that of all clouds (Figure 7a). High cloud amounts in April and December

appear above 1 km; while not all of them can be detected by CloudSat, these clouds can be detected by CALIPSO due to the low cloud optical thickness above. The combined CloudSat and CALIPSO detects most of the clouds that surface observations see except below 1 km (Figure 7b, 7c). Liquid clouds are most common in the lowest 1 km, mainly from April to September; they also appear in higher altitudes from June to September, e.g. around 9% at 2750 m in September (Figure 7d). The combined CloudSat and CALIPSO detects most liquid clouds above 1 km, with the differences from the surface observations less than

1.0% in most months. The major negative differences are below 1 km from March to October (Figure 7e, 7f). The mixed phase clouds are most common in the lowest 1 km, mainly in August and September, but also appearing in higher altitudes from June to September (Figure 7g). The combined CloudSat and CALIPSO detects most mixed phase clouds above 1 km, with the differences from surface observations less than 0.5% in all months. The major negative differences are below 600 km from May to October (Figure 7h, 7i). These features are reflected in the mean cloud amount vertical profiles for ice clouds, liquid

clouds, and mixed phase clouds, respectively (Figure 9, Table 3).

- *Table 3: Mean cloud amount from surface observations, CALIPSO, CloudSat, and combined CALIPSO and CloudSat (CC) for ice clouds, liquid clouds, and mixed phase clouds.*

| Height (m) | Ice clouds (%) | | | | Liquid clouds (%) | | | | Mixed phase clouds | | | |
|---|---|---|---|---|---|---|---|---|---|---|---|---|
| | Surface | Calipso | CloudSat | CC | Surface | Calipso | Cloudsat | CC | Surface | Calipso | CloudSat | CC |
| 149.5 | 37.7 | 21.0 | 0.0 | 21.0 | 9.9 | 2.2 | 0.0 | 2.2 | 12.1 | 5.7 | 0.0 | 5.7 |
| 275.5 | 31.8 | 17.9 | 0.0 | 17.9 | 11.8 | 4.4 | 0.0 | 4.4 | 14.3 | 7.3 | 0.0 | 7.3 |
| 401.5 | 27.2 | 15.2 | 0.0 | 15.2 | 12.1 | 6.2 | 0.0 | 6.2 | 12.3 | 6.3 | 0.0 | 6.3 |
| 527.5 | 24.0 | 13.2 | 0.0 | 13.2 | 9.4 | 4.7 | 0.0 | 4.7 | 9.5 | 4.7 | 0.0 | 4.7 |
| 653.5 | 22.1 | 12.4 | 1.4 | 13.6 | 8.5 | 4.7 | 0.0 | 4.7 | 8.4 | 4.1 | 0.4 | 4.5 |
| 779.5 | 20.7 | 11.8 | 7.9 | 16.0 | 8.6 | 4.9 | 0.9 | 5.4 | 7.9 | 3.8 | 4.3 | 6.2 |
| 905.5 | 18.8 | 10.8 | 11.9 | 16.2 | 6.8 | 3.3 | 3.7 | 5.5 | 6.3 | 3.1 | 5.7 | 5.9 |
| 1050.0 | 18.1 | 10.6 | 13.1 | 16.2 | 6.2 | 3.2 | 4.2 | 5.5 | 6.1 | 3.1 | 5.7 | 5.9 |
| 1250.0 | 16.8 | 10.2 | 12.6 | 15.3 | 3.9 | 1.7 | 3.0 | 3.4 | 4.9 | 2.2 | 4.7 | 4.7 |
| 1450.0 | 16.1 | 10.0 | 12.5 | 14.9 | 3.8 | 1.6 | 2.8 | 3.3 | 4.3 | 2.0 | 4.1 | 4.2 |
| 1650.0 | 16.0 | 10.2 | 12.6 | 15.0 | 4.3 | 1.9 | 3.1 | 3.8 | 3.8 | 1.7 | 3.7 | 3.8 |
| 1850.0 | 15.9 | 10.5 | 12.6 | 15.0 | 3.9 | 1.8 | 2.8 | 3.5 | 3.6 | 1.5 | 3.5 | 3.5 |
| 2050.0 | 16.1 | 11.0 | 12.7 | 15.2 | 2.9 | 1.2 | 2.1 | 2.5 | 3.4 | 1.3 | 3.3 | 3.4 |





| 2550.0 | 15.1 | 11.5 | 12.4 | 14.7 | 2.9 | 1.1 | 2.3 | 2.6 | 3.5 | 1.6 | 3.4 | 3.5 |
|--------|------|------|------|------|-----|-----|-----|-----|-----|-----|-----|-----|
| 3050.0 | 16.4 | 13.4 | 13.0 | 16.0 | 1.8 | 0.9 | 1.3 | 1.6 | 2.9 | 1.7 | 2.8 | 2.8 |
| 3550.0 | 16.9 | 14.7 | 13.4 | 16.7 | 1.4 | 0.7 | 1.0 | 1.3 | 2.5 | 1.6 | 2.5 | 2.5 |
| 4050.0 | 17.0 | 15.6 | 13.3 | 16.9 | 0.9 | 0.5 | 0.7 | 0.9 | 2.3 | 1.5 | 2.2 | 2.3 |
| 5050.0 | 17.1 | 16.4 | 13.5 | 17.1 | 0.9 | 0.6 | 0.7 | 0.9 | 1.3 | 1.0 | 1.3 | 1.3 |
| 6050.0 | 16.0 | 15.8 | 11.8 | 16.0 | 0.2 | 0.1 | 0.1 | 0.2 | 0.8 | 0.7 | 0.7 | 0.8 |
| 7050.0 | 12.1 | 12.1 | 7.9  | 12.1 | 0.0 | 0.0 | 0.0 | 0.0 | 0.4 | 0.4 | 0.4 | 0.4 |
| 8050.0 | 7.8  | 7.8  | 4.6  | 7.8  | 0.0 | 0.0 | 0.0 | 0.0 | 0.1 | 0.1 | 0.1 | 0.1 |
| 9050.0 | 3.2  | 3.2  | 1.5  | 3.2  | 0.0 | 0.0 | 0.0 | 0.0 | 0.0 | 0.0 | 0.0 | 0.0 |
| 10050.0 | 0.7 | 0.7 | 0.3  | 0.7  | 0.0 | 0.0 | 0.0 | 0.0 | 0.0 | 0.0 | 0.0 | 0.0 |
| 11050.0 | 0.1 | 0.1 | 0.0  | 0.1  | 0.0 | 0.0 | 0.0 | 0.0 | 0.0 | 0.0 | 0.0 | 0.0 |
| 12050.0 | 0.0 | 0.0 | 0.0  | 0.0  | 0.0 | 0.0 | 0.0 | 0.0 | 0.0 | 0.0 | 0.0 | 0.0 |

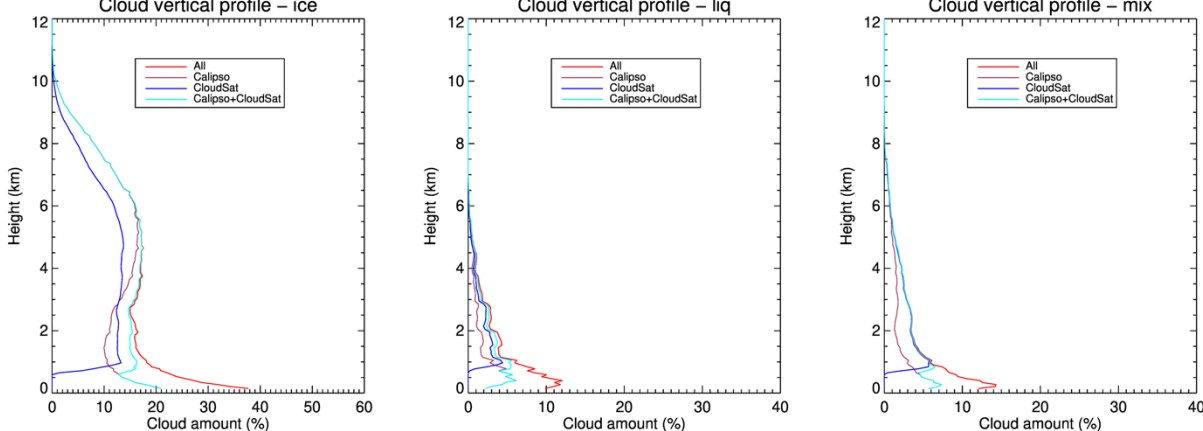

Figure 9: Mean cloud amount vertical distributions from surface observations and from simulated CloudSat, CALIPSO, and combined CloudSat and CALIPSO for (a) ice (b) liquid and (c) mixed phase cloud during the Surface Heat Budget of the Arctic Ocean (SHEBA) experiment.

The mean cloud vertical profiles for all, ice, liquid, and mixed phase clouds in the winter months (November to March, Figure S3a-d) and the summer months (May to September, Figure S3e-h) show similar features as those for all months (Figure 8, 9). Mean overall cloud amounts in the summer months have higher values at all levels, especially below 1 km mainly due to higher cloud liquid and mixed phase clouds. The CloudSat mean cloud amounts above 1 km are smaller than those from the surface observations with relatively constant differences between 3.0-4.0% due to not detecting thin ice clouds. The differences



between CALIPSO and the surface cloud amount increase with decreasing altitude due to the increasing accumulated cloud optical thickness. The combined CloudSat and CALIPSO detects most of the clouds that surface observations see above 1 km, but much fewer below 1 km mainly due to the surface clutter effect of CloudSat and limited CALIPSO detection capability below 1 km. Because of more liquid clouds, mixed phase clouds, and even ice clouds in higher altitudes and the associated

higher accumulated cloud optical thickness above 1 km in the summer months than in the winter months, the differences in cloud amount between the combined CloudSat and CALIPSO and the surface observations are larger in summer months than in winter months.

## 3.2 Uncertainty in Cloud Radiative Forcing due to cloud detection limitations near the surface

### 3.2.1 Surface

During the SHEBA experiment, CRFs, computed with complete cloud profiles, are positive most of the year, except around 40 days centered on Julian day 560 starting from January 1, 1997 (Figure 10). As a result, the monthly mean CRFs are positive in all months expect in July (Figure S4, Table 4). This result suggests that clouds warm the surface for most of the year except a short time in July over the Arctic Ocean, which is consistent with the conclusions of previous studies (Intrieri et al. 2002a, Shupe and Intrieri 2004). There are two major components of the CRF, longwave and shortwave CRFs. The longwave CRFs

are positive all year with the maximum in September and the minimum in February. The shortwave CRFs are zero from October to February due to little incoming shortwave radiation, and are negative in other months. The maximum negative values of the shortwave CRF are in July 1998 (Figure S4, Table 4), which can be attributed to the combination of the near maximum accumulated TOA incoming shortwave radiation flux and the minimum surface albedo in July during the SHEBA experiment (Letterly et al. 2018, Shupe and Intrieri 2004).

▪ Table 4: Monthly mean cloud radiative forcing (CRF) at the surface for longwave (LW), shortwave (SW), and the combined LW and SW (all) with the clouds from the surface observations collected during the Surface Heat Budget of the Arctic Ocean (SHEBA) experiment and the differences between the CRF with clouds in the surface observations only identified from combined CloudSat and CALIPSO, CALIPSO, or CloudSat and the CRF from the clouds from the surface observations.

| | All clouds from surface observations | | | (CloudSat+calipso)- clouds from surface | | | Calipso-clouds from surface | | | CloudSat-clouds from surface | | |
|---|---|---|---|---|---|---|---|---|---|---|---|---|
| | LW | SW | all | LW | SW | all | LW | SW | all | LW | SW | all |
| Oct | 32.7 | -0.1 | 32.6 | -1.0 | 0.0 | -1.0 | -2.1 | 0.0 | -2.1 | -14.2 | 0.1 | -14.1 |
| Nov | 34.2 | 0.0 | 34.2 | -0.2 | 0.0 | -0.2 | -2.4 | 0.0 | -2.4 | -9.0 | 0.0 | -9.0 |
| Dec | 21.0 | 0.0 | 21.0 | 0.2 | 0.0 | 0.2 | -0.2 | 0.0 | -0.2 | -2.7 | 0.0 | -2.7 |
| Jan | 22.0 | 0.0 | 22.0 | 0.3 | 0.0 | 0.3 | 0.1 | 0.0 | 0.1 | -9.1 | 0.0 | -9.1 |
| Feb | 20.5 | -0.2 | 20.3 | 0.2 | 0.0 | 0.2 | -0.4 | 0.0 | -0.4 | -3.1 | 0.0 | -3.1 |



| Mar | 34.6 | -4.2 | 30.4 | 0.1 | 0.3 | 0.4 | -2.4 | 0.5 | -1.8 | -8.0 | 1.2 | -6.8 |
|---|---|---|---|---|---|---|---|---|---|---|---|---|
| Aprl | 42.5 | -12.6 | 29.9 | -0.9 | 0.6 | -0.3 | -3.1 | 1.6 | -1.5 | -15.9 | 4.9 | -11.0 |
| May | 43.3 | -22.3 | 21.1 | -3.0 | 2.9 | -0.1 | -5.0 | 4.4 | -0.6 | -22.6 | 11.5 | -11.1 |
| Jun | 44.4 | -34.5 | 10.0 | -2.5 | 3.6 | 1.1 | -9.1 | 10.0 | 0.9 | -16.4 | 11.1 | -5.3 |
| Jul | 43.9 | -61.0 | -17.1 | -3.4 | 6.1 | 2.7 | -9.1 | 17.6 | 8.6 | -15.7 | 19.0 | 3.3 |
| Aug | 59.8 | -33.9 | 25.9 | -2.8 | 4.0 | 1.2 | -10.0 | 9.4 | -0.7 | -20.9 | 13.0 | -7.9 |
| Sept | 63.2 | -8.2 | 55.0 | -3.0 | 0.4 | -2.6 | -15.0 | 2.6 | -12.4 | -15.3 | 1..9 | -13.5 |

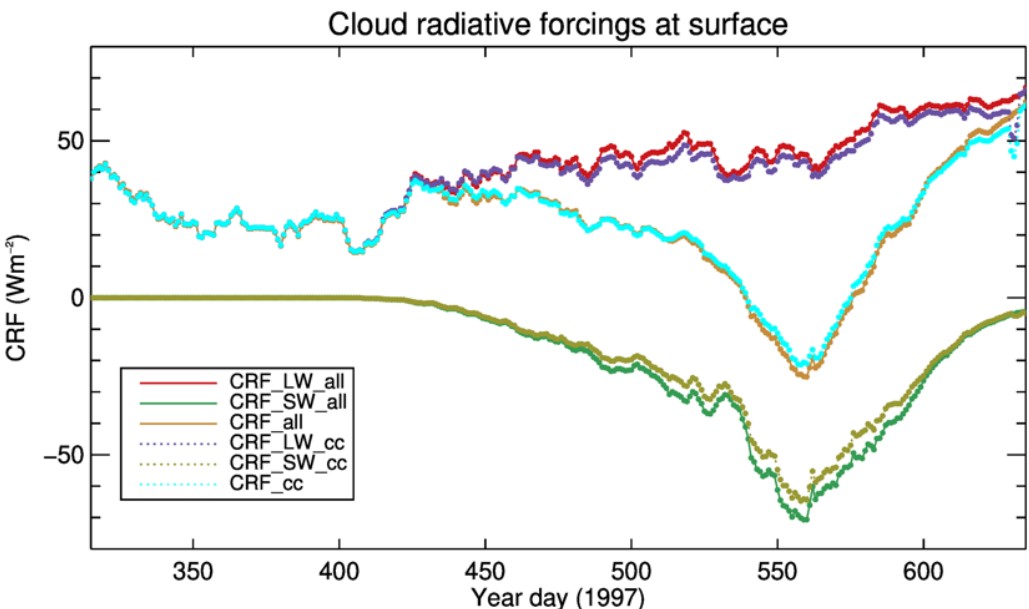

Figure 10: Cloud radiative forcing (CRF) with a 20-day running average of daily means at the surface for longwave (LW) and shortwave (SW) with clouds from surface observations collected during the Surface Heat Budget of the Arctic Ocean (SHEBA) experiment (all) and CRF with clouds identified by the combined CloudSat and CALIPSO (cc). The Julian day starts from January 1, 1997.

CRFs computed with cloud profiles identified by the combined CloudSat and CALIPSO cloud mask show a very similar annual cycle and values to those using complete cloud profiles (Figure 10, Table 4). Their absolute differences are less than 3 $Wm^{-2}$ for all months, with a maximum of 2.7 $Wm^{-2}$ in July. The differences in their longwave CRFs are mostly near zero in




the winter months, with larger negative values from May to September (Figure S4, Table 4). The differences in their shortwave CRF are positive and larger in the summer months, but zero in the winter months. The cancellation of the shortwave and longwave CRF differences in the summer months and both differences being near zero in the winter months lead to smaller CRF differences.

Though the differences in monthly mean CRFs are small, large ranges in both the longwave and the shortwave CRFs exist in individual values, calculated using the individual 1-min-interpolated profiles. The longwave CRFs at the surface in the winter months, November to March, mostly have absolute differences within 2 Wm⁻², with limited cases outside that range (Figure 11b); the same conclusion holds for the cases in which the clouds near the surface not detected by the combined CloudSat and CALIPSO are ice, liquid, or mixed phase clouds (Figure S5). In the summer months, the differences in the longwave CRFs at

the surface show mostly negative values, as large as -30 Wm⁻² (Figure 11). These differences are more negative if liquid or mixed phase clouds are not detected near the surface (Figure S6). The differences in the shortwave CRFs are all positive, as large as 30 Wm⁻², in the summer months (Figure 12a), and are small (large) if ice (liquid, or mixed phase) clouds are not detected near surface (Figure 12b, 12c, 12d).

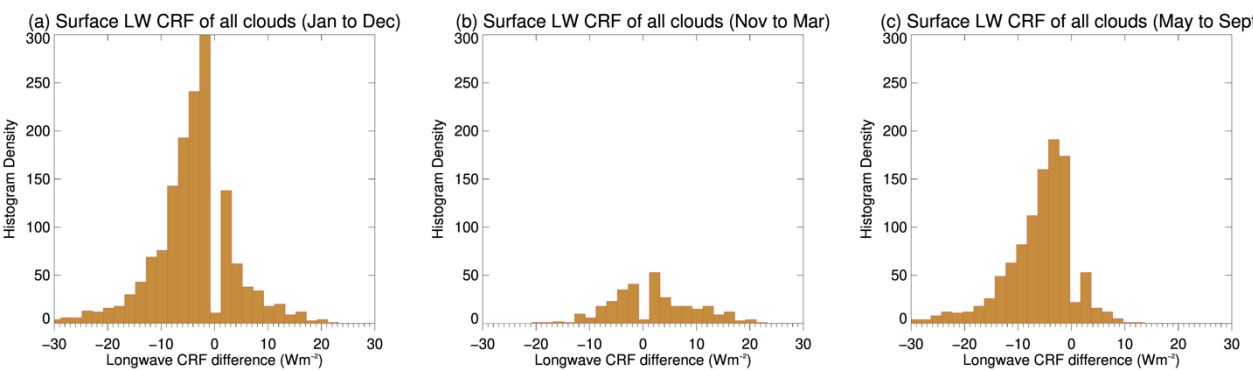

Figure 11: Histogram of longwave cloud radiative forcing at the surface between using clouds identified by combined CloudSat and CALIPSO and using clouds from surface observations during the Surface Heat Budget of the Arctic Ocean (SHEBA) experiment for (a) January to December, (b) November to March, and (c) May to September. Cases with absolute differences less than 2 Wm⁻² are the majority while excluded in the histogram.



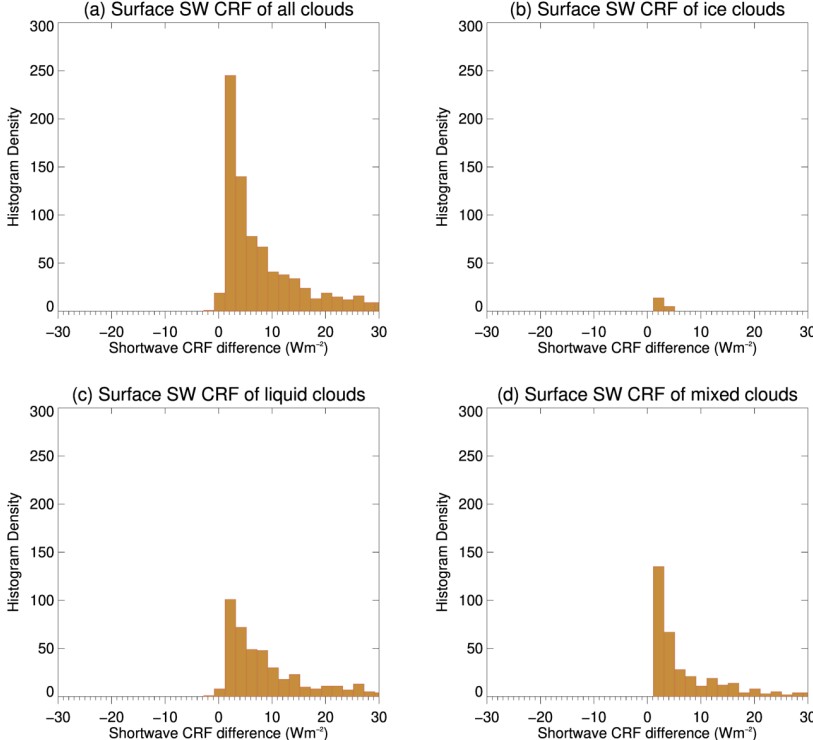

Figure 12: Histogram of shortwave cloud radiative forcing at surface between using clouds identified by combined CloudSat and CALIPSO and clouds from surface observations during the Surface Heat Budget of the Arctic Ocean (SHEBA) experiment for (a) all clouds, (b) ice clouds, (c) liquid clouds and (d) mixed phase clouds.

When the cloud profiles determined from only the CloudSat cloud mask are used, the differences in the CRFs are much larger than the differences using the cloud profiles determined from the combined CloudSat and CALIPSO (Table 4), e.g. -14.1 Wm$^{-2}$ in October and -13.5 Wm$^{-2}$ in September. The differences in the longwave and shortwave CRFs are even larger, which cancel each other somewhat for smaller CRFs. When the cloud profiles determined from only the CALIPSO cloud mask are available,

10   the differences in the CRFs are smaller than the differences with cloud profiles from only the CloudSat and larger than the differences with the combined CloudSat and CALIPSO. For example, the differences are 8.6 Wm$^{-2}$ in July and -12.4 Wm$^{-2}$ in September. The differences in the longwave CRFs have large negative values and the differences in shortwave CRFs have large positive values, which cancel each other for smaller CRFs (Table 4).

The longwave CRF differences at the surface are close to zero in the winter months, and tend more towards large negative

15   values in the summer months especially when omitting the liquid and mixed phase clouds near the surface (Figure 11, A6). In the summer months, the longwave CRF differences become more negative with the increasing optical thickness of those clouds below 1 km and therefore not detected by the combined CloudSat and CALIPSO (Figure 13). Because of the temperature inversions near the surface in the winter months, clouds in the lowest 1 km can have temperatures higher than, equal to, or





lower than the surface temperature, though the cloud temperatures are not far from the surface temperature (Figure 14a). Removal of these clouds would not likely greatly change the downward longwave radiation at the surface, which leads to the small longwave CRF differences in the winter months. While in the summer months, the temperature inversions are not as common; thus the clouds in the lowest 1 km often have temperatures lower than the surface, with even lower cloud

5    temperatures above 1km (Figure 14b). Removal of more clouds near the surface would lead to less downward longwave radiation, thus smaller longwave CRFs, and larger negative differences.

The SW CRF differences at the surface are positive in the summer months especially when omitting liquid and mixed phase clouds near the surface (Figure 12). Larger positive SW CRF differences are associated with increasing optical thickness of those clouds below 1 km and not detected by the combined CloudSat and CALIPSO (Figure 15). In the summer months,

10    especially in June, July, and August, the surface albedo decreases due to the melt ponds and more open water; the surface absorbs more shortwave radiation without clouds above. Clouds especially those that are liquid and mixed phase are brighter than the surface and reflect more shortwave radiation back to space, thus leading to smaller downward shortwave radiation at the surface and larger negative shortwave CRFs. Removal of more clouds near the surface would lead to larger downward shortwave radiation, smaller negative shortwave CRFs, and larger positive differences.

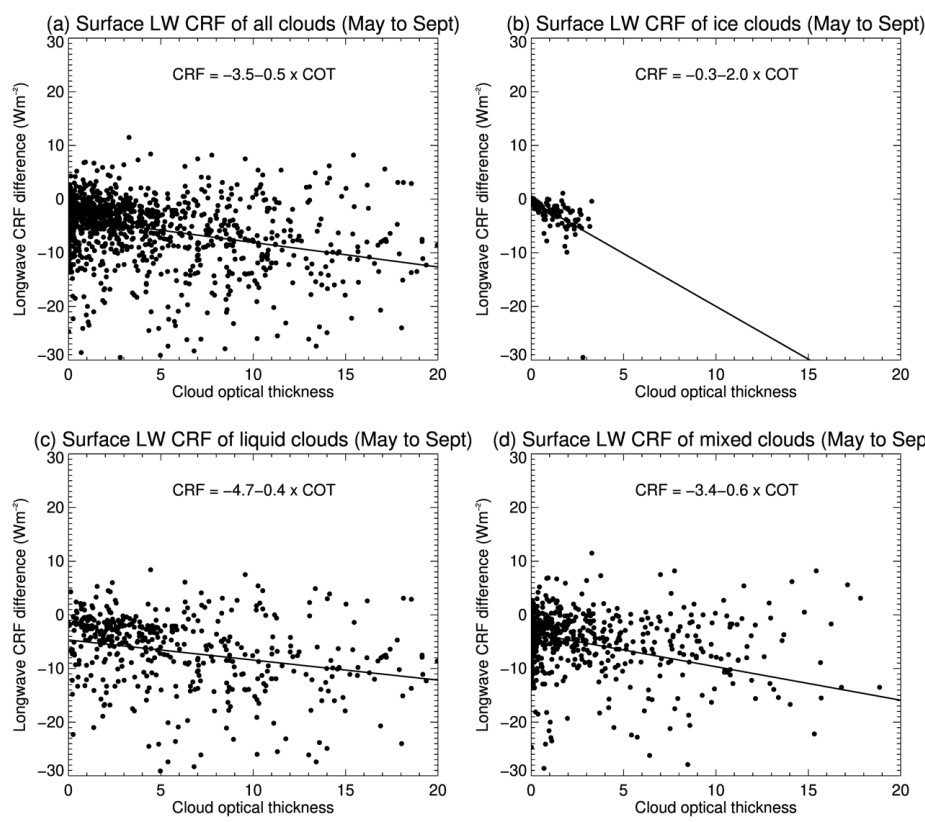





Figure 13: Scattering plots of longwave cloud radiative forcing at the surface between using clouds identified by combined CloudSat and CALIPSO and clouds from surface observations and cloud optical thickness for (a) all clouds, (b) ice clouds, (c) liquid clouds, and (d) mixed phase clouds below 1 km from May to September during the Surface Heat Budget of the Arctic Ocean (SHEBA) experiment.

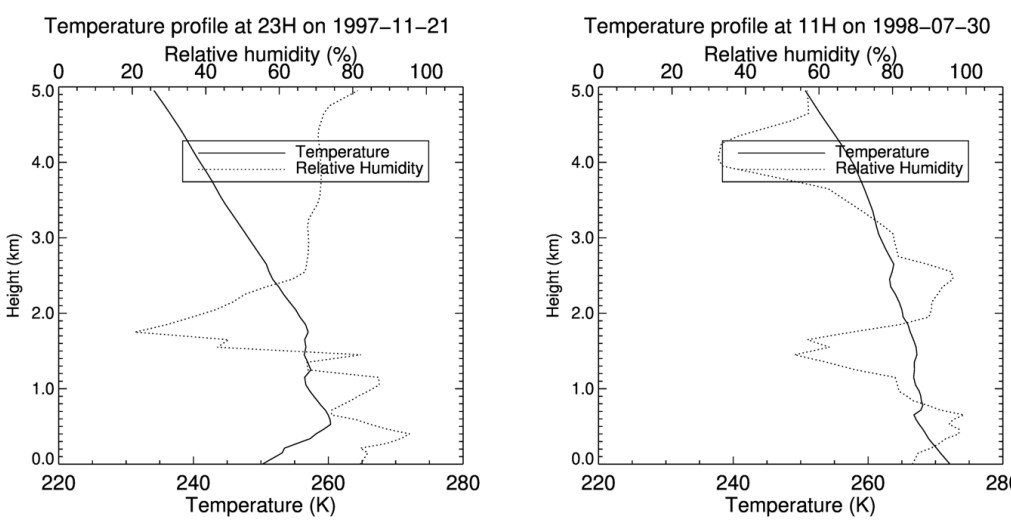

Figure 14: Temperature and relative humidity profiles on November 21, 1997 and on July 30, 1998 over the Arctic Ocean during the Surface Heat Budget of the Arctic Ocean (SHEBA) experiment.

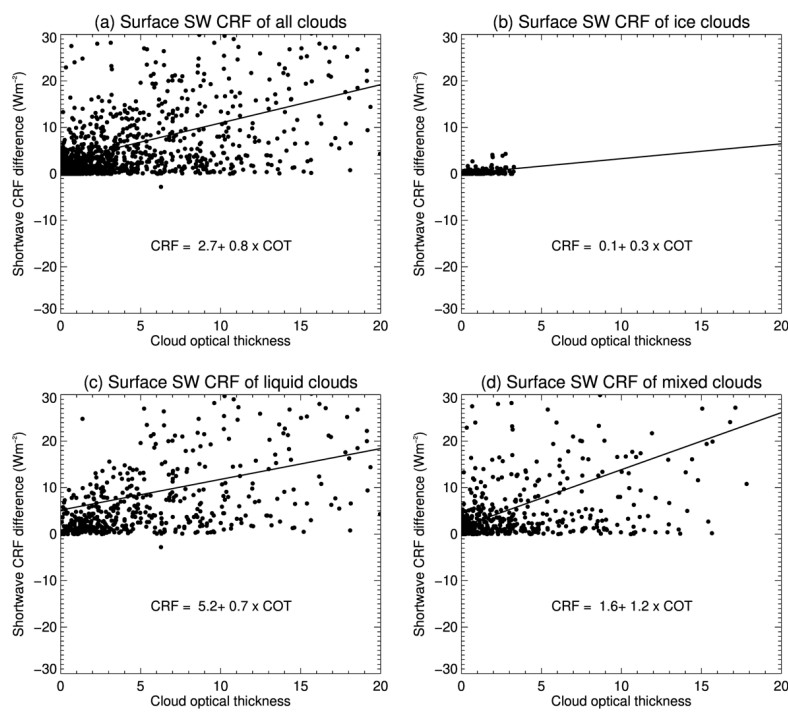



Figure 15: Scattering plots of shortwave cloud radiative forcing at the surface between using clouds from surface observations and clouds identified by combined CloudSat and CALIPSO and cloud optical thickness of omitted (a) all clouds, (b) ice clouds, (c) liquid clouds, and (d) mixed phase clouds below 1 km during the Surface Heat Budget of the Arctic Ocean (SHEBA) experiment.

**3.2.2 Top of atmosphere (TOA)**

CRFs at TOA are negative from April to August, and positive from September to March during the SHEBA experiment (Figure 16, Table 5, Figure S7). The absolute of the negative values is much larger than the absolute of the positive values. This suggests the clouds cool the Arctic Ocean in the summer months, except September, and slightly warm the Earth in the winter months, with an overall impact of cooling the Arctic Ocean. The bright clouds in the summer months reflect more shortwave radiation back to space than even the snow-covered surface especially in July when the surface albedo is low due to melt ponds and more open water, such that the shortwave CRFs at TOA are all negative in the summer months. On the other side, the longwave CRFs at TOA are positive in all months possibly due to colder effective cloud top temperature than the surface even in the winter months when surface temperature inversions are common. The shortwave and longwave CRFs somewhat cancel each other in most months.

▪ Table 5: Monthly mean cloud radiative forcing (CRF) at the top of atmosphere for longwave (LW), shortwave (SW), and the combined LW and SW (all) with the clouds from the surface observations collected during the Surface Heat Budget of the Arctic Ocean (SHEBA) experiment and the differences between the CRF with clouds in the surface observations only identified from combined CloudSat and CALIPSO, CALIPSO, or CloudSat and the CRF from the clouds from the surface observations.

| | All clouds from surface observations | | | (CloudSat+calipso)-clouds from surface | | | Calipso-clouds from surface | | | CloudSat-clouds from surface | | |
|---|---|---|---|---|---|---|---|---|---|---|---|---|
| | LW | SW | all | LW | SW | all | LW | SW | all | LW | SW | all |
| Oct | 3.1 | -0.2 | 2.9 | 0.0 | 0.0 | 0.0 | -0.1 | 0.0 | -0.1 | 0.0 | 0.0 | 0.1 |
| Nov | 5.7 | 0.0 | 5.7 | 0.0 | 0.0 | 0.0 | -0.1 | 0.0 | -0.1 | 0.3 | 0.0 | 0.3 |
| Dec | 1.2 | 0.0 | 1.2 | 0.0 | 0.0 | 0.0 | 0.0 | 0.0 | 0.0 | 0.2 | 0.0 | 0.2 |
| Jan | 1.5 | 0.0 | 1.5 | 00 | 0.0 | 0.0 | 0.0 | 0.0 | 0.0 | 0.7 | 0.0 | 0.7 |
| Feb | 1.9 | -0.3 | 1.6 | 0.0 | 0.0 | 0.0 | 0.0 | 0.0 | 0.0 | 0.3 | 0.0 | 0.3 |
| Mar | 4.9 | -4.3 | 0.7 | -0.1 | 0.2 | 0.0 | -0.1 | 0.3 | 0.1 | -0.4 | 0.8 | 0.4 |
| April | 5.6 | -9.6 | -4.1 | -0.1 | 0.3 | 0.2 | -0.2 | 0.9 | 0.7 | -0.3 | 3.0 | 2.7 |
| May | 10.2 | -16.6 | -6.5 | -0.3 | 1.5 | 1.1 | -0.6 | 2.5 | 1.9 | -1.9 | 7.2 | 5.3 |
| Jun | 16.9 | -30.1 | -13.2 | -0.5 | 2.1 | 1.6 | -1.5 | 7.0 | 5.5 | -1.5 | 7.3 | 5.8 |
| Jul | 20.2 | -59.2 | -39.0 | -0.7 | 4.7 | 4.0 | -1.7 | 15.1 | 13.5 | -1.1 | 15.3 | 14.2 |





| Aug | 15.1 | -30.9 | -15.7 | 0.0 | 2.7 | 2.6 | -0.8 | 6.7 | 5.9 | -0.8 | 9.7 | 8.8 |
| Sept | 18.6 | -8.4 | 10.2 | 0.0 | 0.2 | 0.1 | -2.1 | 1.9 | -0.1 | -0.2 | 1.2 | 1.0 |

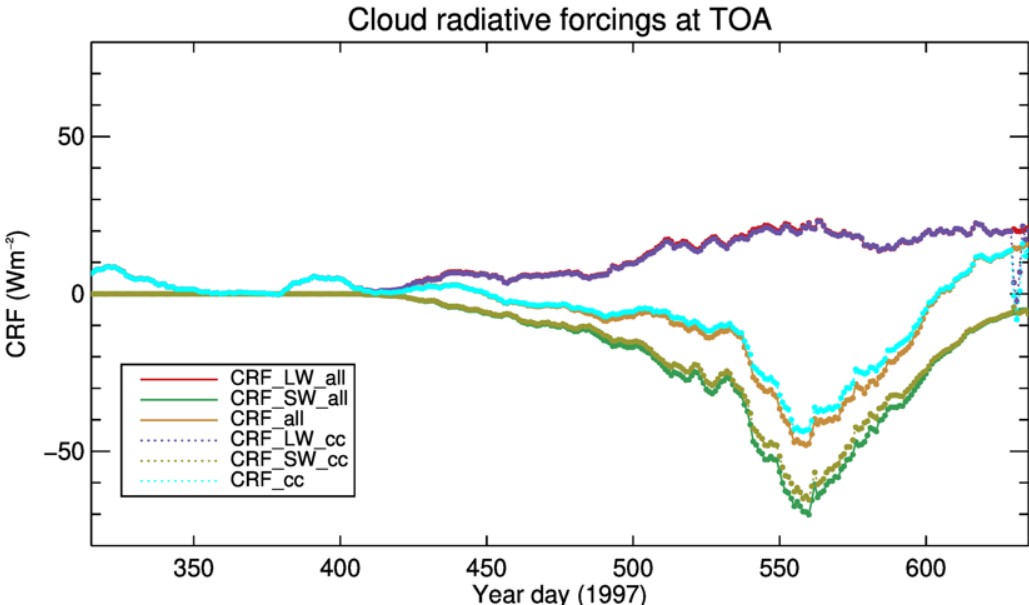

Figure 16: Cloud radiative forcing (CRF) with a 20-day running average of daily means at the top of atmosphere (TOA) for longwave (LW) and shortwave (SW) with clouds from surface observations collected during the Surface Heat Budget of the Arctic Ocean (SHEBA) experiment (all) and CRF with clouds identified by the combined CloudSat and CALIPSO (cc). The Julian day starts from January 1, 1997.

For the impact of cloud detection limitation near the surface by the combined CloudSat and CALIPSO, the CRF differences are positive in all months, with the maximum monthly mean difference at 4.0 $Wm^{-2}$ in July. The differences in the longwave CRFs are near zero in all months, with the maximum magnitude in July at -0.7 $Wm^{-2}$. The differences in the shortwave CRFs are all positive with larger values than the longwave CRF differences, with the maximum magnitude in July at 4.7 $Wm^{-2}$. The CRFs are determined by the shortwave CRFs because of the much smaller longwave CRFs. The differences in shortwave CRFs are all positive and in the summer months (Figure 17a), and these differences are larger if liquid, or mixed phase, clouds are not detected near the surface (Figure 17b, 17c). Few cases exist when only ice clouds are not detected near the surface.



When cloud profiles from only the CloudSat cloud mask are available, the differences in the CRFs from CloudSat become much larger than the differences using the combined CloudSat and CALIPSO (Table 5), e.g. 14.2 Wm$^{-2}$ in July for only CloudSat compared to 4.0 Wm$^{-2}$ in July for the combined CloudSat and CALIPSO. When cloud profiles from only the CALIPSO cloud mask are available, the differences in CRFs from CALIPSO are smaller than those from only CloudSat and larger than those from the combined CloudSat and CALIPSO, e.g. 13.5 Wm$^{-2}$ for CALIPSO in July. For both, the differences in the longwave CRFs are not affected much, while the differences in the shortwave CRFs become large positive values. These changes in the shortwave CRFs determine the CRFs (Table 5).

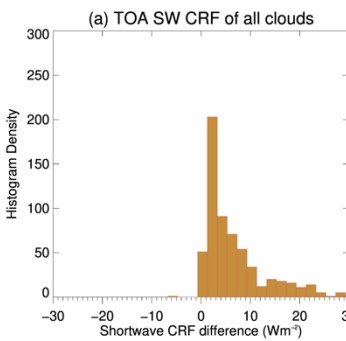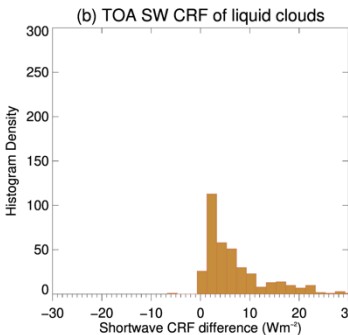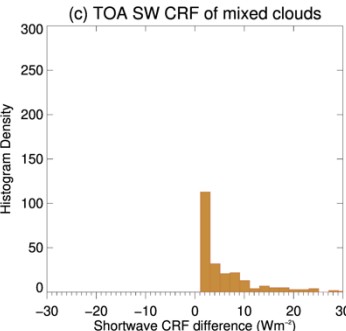

Figure 17: Histogram of shortwave cloud radiative forcing at the top of atmosphere (TOA) between using clouds from surface observations and clouds identified by combined CloudSat and CALIPSO for (a) all clouds, (b) liquid clouds and (c) mixed phase clouds during the Surface Heat Budget of the Arctic Ocean (SHEBA) experiment.


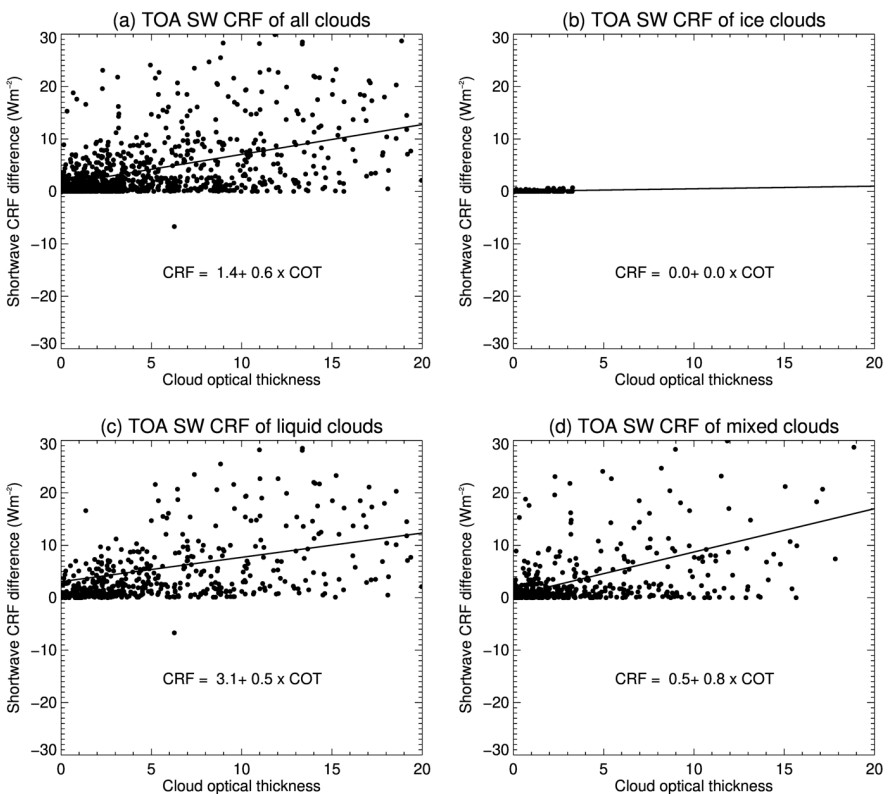

Figure 18: Scattering plots of shortwave cloud radiative forcing at the top of atmosphere (TOA) between using clouds from surface observations and clouds identified by combined CloudSat and CALIPSO and cloud optical thickness of omitted (a) all clouds, (b) ice clouds, (c) liquid clouds, and (d) mixed phase clouds below 1 km during the Surface Heat Budget of the Arctic Ocean (SHEBA) experiment.

The cloud detection limitations near the surface by the combined CloudSat and CALIPSO does not affect the longwave CRFs at TOA, but leads to positive shortwave CRF differences at TOA (Figure 17). These positive differences are larger with increasing optical thickness of those clouds below 1 km and not detected by the combined CloudSat and CALIPSO (Figure 18). In the summer months, the surface albedo decreases, which means the surface reflects less shortwave radiation in clear sky conditions. Clouds especially those that are liquid and mixed phase near the surface help brighten the clouds so that they reflect more shortwave back to space, which leads to larger upward shortwave radiation at the TOA and larger negative shortwave CRFs. Removal of more clouds near the surface would reduce the brightness of the cloud, resulting in smaller upward shortwave radiation at TOA, smaller negative shortwave CRFs, thus larger positive differences.



## 3.3 Uncertainty in the results

The CloudSat and CALIPSO cloud masks are based on the simulated radar reflectivity and cloud optical thickness using the retrieved cloud properties from the surface-based combined radar and lidar observations during the SHEBA experiment. For CloudSat, a vertical profile of fixed thresholds is applied to identify the CloudSat cloud mask, and a layer with radar reflectivity

higher than the threshold at that layer is identified as cloud; for CALIPSO, a layer with cloud optical thickness larger than 0 and an accumulated cloud optical thickness from TOA to the layer above this layer less than a fixed threshold, i.e. 5, is identified as cloud. There are also uncertainties in the cloud property retrievals. In addition, the surface-based radar and lidar observations are usually reasonable only at heights greater than 100 -150 m above ground (Griesche et al. 2021, Hu et al. 2021). However, these profiles of cloud properties serve as a reasonable data set of possible Arctic cloud scenarios covering a

whole year, which serve as a reasonable data set to conduct this study. Retrieval uncertainties in these cloud properties do not affect the main conclusion of this study.

There are a few factors leading to uncertainties in the CloudSat and CALIPSO cloud masks, and consequently the derived cloud amounts and computed radiation fluxes based on these masks. These factors include the accuracy of the simulated radar reflectivity and the cloud optical thickness, and the cloud detection methods for CloudSat and CALIPSO. It is challenging to

evaluate these uncertainties because of the lack of truth data, e.g. scarcity of the collocated cloud observations from in situ cloud measurements and surface-based and space-based radar-lidar cloud observations. Many studies have shown the capability of QuickBeam to simulate CloudSat reflectivity (Bodas-Salcedo et al. 2011, Zhang et al. 2018). Even though, there was no CloudSat data in 1997 and 1998 to validate the simulated CloudSat reflectivity, there were surface-based MMCR observations at 35 GHz. Instead of validating simulated CloudSat reflectivity, we compared the surface-based MMCR

observations to the simulated radar reflectivity at 35 GHz at the surface. Their differences reflect the combined uncertainties in the cloud property retrievals and the QuickBeam simulations. Such a case is shown in Figure 19. Results show small differences for ice cloud reflectivity and relatively large differences for liquid cloud reflectivity.







Figure 19: (a) Observed 35 GHz millimeter cloud radar (MMCR) reflectivity and (b) difference between the simulated and the observed MMCR reflectivity on November 21, 1997.

5    The CALIPSO cloud detection capability depends on the threshold of the accumulated cloud optical thickness, and different thresholds have been applied in previous research (Hu et al. 2007). A cloud optical thickness threshold of 5 is used in this study. Different thresholds, e.g. 4 and 6, are applied to estimate the CALIPSO cloud detection sensitivity to this threshold. The mean cloud amount vertical profiles with thresholds of 4, 5, and 6 are presented in Figure 20 for all clouds, ice clouds, liquid





clouds, and mixed phase clouds. With a smaller (larger) threshold, the CALIPSO cloud amounts are smaller (larger) because the CALIPSO detects fewer (more) clouds at the lower levels. The cloud amount differences are -3.0% (3.0%) at 149.5 m, -1.7% (1.7%) at 1050 m, and around 0% at 6050 m for all clouds, -2.0 % (2.0%) at 149.5 m, 1.0% (-1.0%) at 1050 m, and around 0% at 6050 m for ice clouds, and -0.5% (0.5%) at 149.5 m and 1050 m, and around 0% at 4050 m for liquid and mixed phase clouds. It should be noted that a more sophisticated detection scheme is used in the operational CALIPSO cloud mask product (Winker et al. 2009), and results in this study should not be treated as the results from operational products.

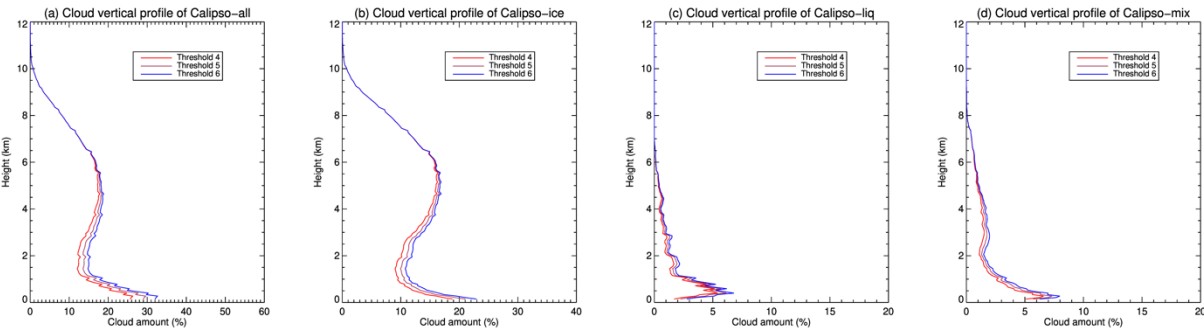

Figure 20: CALIPSO cloud amount vertical profile with three optical thickness thresholds of 4, 5, and 6 for (a) all, (b) ice, (c) liquid and (d) mixed phase clouds.

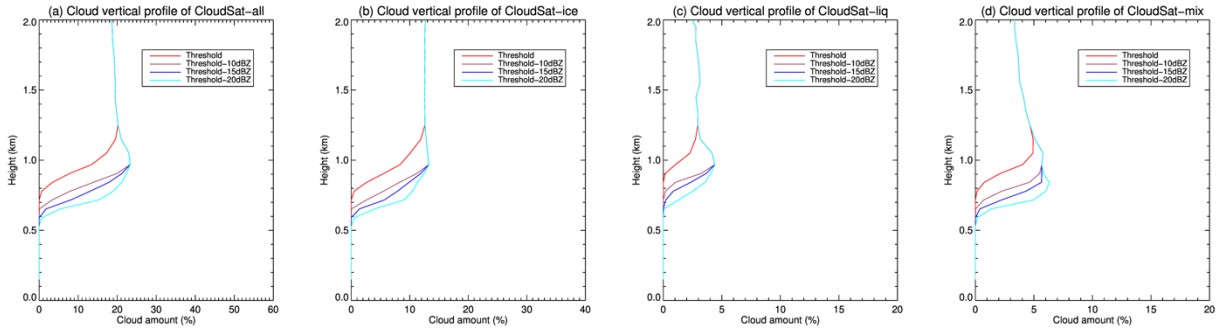

Figure 21: CloudSat cloud amount vertical profile with different vertical detection thresholds for (a) all, (b) ice, (c) liquid and (d) mixed phase clouds.

In a similar way, the CloudSat cloud detection capability depends on the fixed vertical thresholds selected. Smaller or larger thresholds at each layer affect the cloud amount of that layer with the simulated radar reflectivity. Because the focus of this study is on the lower altitudes where the surface clutter impact is strongest, thresholds are changed in the lowest 5 CloudSat range bins (lowest 960 m) to test the sensitivity. The thresholds are set at -10 dBZe, -15 dBZe, and -20 dBZe lower than the mean radar-measured noise power (the 99[th] percentile of the clear-sky returns) while larger than -26 dBZe. The differences in the derived cloud amount are between 550 m and 960 m, with the cloud amounts all 0.0% below 550 m (Figure 21). For the



cloud amount, compared to the results with the threshold of -15 dBZe lower than the mean radar-measured noise power, the differences are -1.5% (3.5%) with offsets of -10 dBZe (-20 dBZe) at 653.5 m, -5.0% (5.0%) from 716.5 m to 905.5 m, and around 0.0% at 968.5 m. It should be noted that a much more sophisticated detection scheme including a few filter schemes is used in the operational CloudSat cloud mask product (Marchand et al. 2008). The detection method in this study is simplified

and should not be used as the results based on operational products.

## 4 Conclusions

Low-level clouds are common in the Arctic. Space-based active lidar and radar with the best detection capability identify 25-40% and 0-25% fewer clouds in absolute values below 0.5 km and between 0.5–1.0 km, respectively, than from surface-based observations over the land area in the Arctic. It is not clear if a similar detection limitation exists over the Arctic Ocean and

how this limitation affects the radiation flux at the surface and TOA. Vertical profiles of cloud classification (cloud phase) and cloud microphysical property retrievals are available from the surface-based lidar and radar observations during the SHEBA experiment. The proxies of CloudSat and CALIPSO cloud masks are derived based on the simulated radar reflectivity and cloud optical thickness using the retrieved cloud properties. Cloud amounts from the CloudSat and CALIPSO cloud masks are compared to those from surface observations to assess the active satellite sensors' low-level cloud detection limitations. Two

sets of cloud profiles are then used to calculate the CRFs at the surface and TOA, one with the complete profiles from the surface observations and the other with only those cloud layers identified by CloudSat and CALIPSO. Differences in these two CRFs are used to quantify the impacts of the active satellite sensors' low-level cloud detection limitations on the CRFs. In our study, the differences/uncertainties in the CRFs mean the differences between the CRFs computed with only cloud layers identified as clouds by the CloudSat, CALIPSO, or the combined CloudSat and CALIPSO and those computed with the

complete cloud profiles from the surface observations. The difference in the cloud amount is the difference between that from the CloudSat, CALIPSO, or the combined CloudSat and CALIPSO cloud mask and that from the surface observations. The primary conclusions are as follows:

- Above 1 km, the combined CloudSat and CALIPSO detects almost all the clouds that surface observations show, except between -8.0% and -3.0% fewer clouds for the clouds between 1 km and 2 km in August and September. The CloudSat
25        or CALIPSO alone detects most of the clouds that the surface observations show with 0% to -5% fewer clouds. Thin ice clouds between 1 km and 8 km in April and December are not detected by CloudSat, and CALIPSO detects much fewer clouds between 1 km and 4.5 km from June to September because of the relatively large amount of liquid and mixed phase clouds. Below 1km, CloudSat detects no clouds below 600 m for all months and the detection gradually increases to around 75% of the clouds the surface sees near 1 km. CALIPSO detects more than half of the clouds the surface sees in
30        the winter months, and around 35% in other summer months. The ratios of the clouds detected from the combined CloudSat and CALIPSO to the cloud amount that surface sees are higher in the winter months than in the summer months, with the highest ratio in December at around 85% and the lowest ratio in September at around 30%.





- For the annual mean cloud amount, the combined CloudSat and CALIPSO detects roughly half of all the clouds that the surface observations see below 600 m, 25% fewer in absolute value. From 600 m to 1 km, the combined CloudSat and CALIPSO cloud amount is 9% less than that from the surface observations. The differences are -2.7% and -1.3% at 1 km and 2 km, respectively. The combined CloudSat and CALIPSO detects most of the clouds that the surface sees above 2 km.

- The liquid and mixed phase clouds are most common in the lowest 1 km. The combined CloudSat and CALIPSO detects the majority of these clouds in the winter months because of the relatively smaller cloud optical thickness above, thus CALIPSO helps in detecting these clouds. While in the summer months, the combined CloudSat and CALIPSO detects a small portion of these clouds.

- The monthly mean CRF uncertainties at the surface due to the cloud detection limitations are small, with a maximum absolute monthly mean value of 2.7 $Wm^{-2}$ in July. In the summer months, the uncertainties in longwave CRFs and shortwave CRFs are large negative and large positive values respectively; they have different signs and cancel each other for small CRFs. In the winter months, the longwave CRF uncertainties at the surface are close to zero, and the shortwave CRFs are 0. In the summer months, the longwave CRF differences become larger negative values and shortwave CRF differences become larger positive values with the increasing optical thickness of those clouds below 1 km and not detected by the combined CloudSat and CALIPSO.

- The uncertainties in the CRFs at TOA are small, with maximum monthly mean values of 4.0 $Wm^{-2}$ in July. The uncertainties in longwave CRFs are close to 0 $Wm^{-2}$ in all months, and the uncertainties in shortwave CRFs determine the CRF uncertainties at TOA. Cloud detection limitation near the surface leads to positive shortwave CRF differences at TOA, and these positive differences are larger with increasing optical thickness for those clouds below 1 km and not detected by the combined CloudSat and CALIPSO.

- Though the monthly mean CRF uncertainties are small, large uncertainty ranges in both longwave and shortwave CRFs exist in individual cases up to 30.0 $Wm^{-2}$.

- Clouds detected by only CALIPSO or CloudSat lead to larger CRF uncertainties with absolute monthly means larger than 10.0 $Wm^{-2}$ for CALIPSO and CloudSat in some months, with even larger shortwave and longwave CRF uncertainties at the surface. At TOA, clouds detected by only CALIPSO or CloudSat also lead to larger shortwave CRF and CRF uncertainties. The CRF uncertainties with only CloudSat are larger than those with only CALIPSO at both surface and TOA.

The findings of the space-based active sensors' cloud detection limitations near the surface over the Arctic Ocean are consistent with previous findings over land in the Arctic. In this study, the approach is different in that we are using simulated CloudSat reflectivity and cloud optical thickness with retrieved cloud properties from surface radar/lidar observations during the SHEBA experiment, while previous studies used collocated CloudSat/CALIPSO and surface-based lidar/radar products of cloud mask and cloud properties. The findings in this study confirm that the combined CloudSat/CALIPSO detects a limited amount of clouds near the surface over different surface types. This result suggests that studies using combined CloudSat/CALIPSO



cloud products over locations where low-level clouds are common need to consider the impacts of omitting them. Low-level clouds are common over the Arctic Ocean and important for the surface radiation balance and sea ice growth/melting. A better approach to detect clouds near the surface from space-based radar and lidar is needed.

The combined CloudSat and CALIPSO reduces the uncertainties in radiation fluxes at the surface and TOA. Uncertainties in the monthly mean radiation flux from the combined CloudSat and CALIPSO have maximum values of 2.7 Wm$^{-2}$ and 4.0 Wm$^{-2}$ at the surface and at the TOA, respectively, which may suggest that the monthly mean radiation fluxes from the combined CloudSat and CALIPSO may be suitable for climate studies. These uncertainties increase with larger cloud optical thickness under 1 km that is not identified by combined CloudSat and CALIPSO. With the recent increase in low-level clouds with more open water in the summer and autumn (Kay and Gettleman 2009, Liu et al. 2012b), these uncertainties may become larger.

Larger uncertainties exist for individual cases, thus uncertainties in radiative fluxes from combined CloudSat and CALIPSO needs to be considered. Also, radiation fluxes computed/retrieved from only CALIPSO or only CloudSat lead to larger uncertainties. This result supports the combined use of CloudSat and CALIPSO data for cloud detection, cloud property retrievals, and radiation flux calculations. This also suggests that larger uncertainties need to be considered for radiation flux products from only CALIPSO or only CloudSat. Even with combined CloudSat and CALIPSO, large uncertainties in radiation

flux are possible.

In this study, we use simple methods to derive the CloudSat cloud mask, CALIPSO cloud mask, and their combined cloud mask. More sophisticated detection schemes are used in the operational CloudSat and CALIPSO cloud mask products. Results in this study need to be confirmed when collocated CloudSat and CALIPSO products with true cloud and radiation flux observations become available (Tjernström et al. 2014, Di Biagio et al. 2020, Blanchard et al. 2021). Also, results are drawn

based on one year's data during the SHEBA experiment. When more cloud data from surface observations are available from field campaigns, e.g. the Multidisciplinary drifting Observatory for the Study of Arctic Climate (MOSAiC) expedition (Shupe et al. 2022), this work needs to be revisited.

### Acknowledgements

This work is supported by the NOAA JPSS Program Office and GOES-R Series Program Office. The author thanks the NOAA

Physical Sciences Laboratory (PSL) for providing all the SHEBA data sets in this study. The author thanks Dr. Matthew Shupe and his group for developing, retrieving, and sharing the cloud property data sets from SHEBA. This study would not be possible without the observations from SHEBA and retrieved products from Dr. Shupe's group. The author thanks Dr. John Haynes for help with using QuickBeam. The author thanks Dr. Jeff Key for help with using Streamer and valuable discussions on this work. The views, opinions, and findings contained in this report are those of the author(s) and should not be construed

as an official National Oceanic and Atmospheric Administration or U.S. Government position, policy, or decision.



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
