# Peer review of "Impacts of Active Satellite Sensors' Low-level Cloud Detection Limitations on Cloud Radiative Forcing in the Arctic"

_Atmospheric Chemistry and Physics, 2022_

## Author Comment (AC1)

Response to Reviewer's Comments:
(*Authors' responses are in italic and in blue color*)

Reviewer 1:

This is an interesting application of GV data to understand how satellite data cloud masks could lead to uncertainty in radiation forcing estimates. The author develops an application of the Quickbeam simulator to the high-temporal resolution ground validation. Uncertainties in the radiative fluxes due to the missed detection of clouds from simulated CloudSat and CALIPSO cloud masks were estimated. They found that the seasonal cycle has a large impact on the magnitude of CRF biases due to changes in cloud microphysics, temperature structure, and surface albedo characteristics. For monthly averages, the combined CloudSat-CALIPSO mask provides the best match to surface observations, however, errors can get large when comparing 1-minute case data. Uncertainties in the cloud detection methods are described in the final section. This work is useful for the community as uncertainty estimates in Arctic regions are needed; the work, however, has numerous issues that need to be tackled first before publication.

*I appreciate the reviewer for the careful review, positive comments, and constructive suggestions. I believe the manuscript has improved greatly with the revision based on those suggestions.*

More significant comments:

1.

Results in Table 4, Table 5, Figure 10, and Figure 6 are difficult to interpret because of the sampling strategy presented. The author mentions that many cloud types are omitted from the study (snow, drizzle, liquid cloud+drizzle, rain, haze, or uncertain retrievals). It is important to understand how often these cases occur and how their exclusion impacts the results presented here. The CRF results are needed, however, if the sampling represents a small fraction of the total CRF they would not be representative of the total population of clouds. It would be helpful to understand the frequency of occurrence of each type mentioned here to get an idea of what is being sampled.

*The focus of study is the impact on the cloud detection and cloud radiative forcing. There might also be larger uncertainties associated with the retrievals including snow, drizzle, liquid cloud+drizzle, rain, or haze, and the radiative transfer models also have higher uncertainties to simulate the reflectivity and radiative fluxes for these cases. For these reasons, I excluded vertical profiles including snow, drizzle, liquid cloud+drizzle, rain, haze, or uncertain retrievals. This might be a good topic for future work.*

*In response to reviewer's suggestion, I provided the percentages of these profiles to all profiles for each month to show how many cases have been excluded. Also, in the discussion section of the revision, I emphasized this limitation and its possible impact on the total CRF "The study focuses on the impacts of active satellite sensors' low-level cloud detection limitations on cloud radiative forcing, so vertical profiles including snow, drizzle, liquid cloud+drizzle, rain, haze, or uncertain retrievals were excluded in calculating the CRFs. There are over 30,000 profiles in every month from October 1997 to September 1998, except that the total profile numbers are around 15,800 in October, which includes October 1997 and October 1998. Of all the profiles in every month from October 1997 to September 1998, the profiles with snow, drizzle, liquid cloud+drizzle, rain, haze, or uncertain retrievals account for 11.6%, 17.0%, 7.3%, 9.0%, 7.3%, 10.5%, 9.1%, 4.9%, 10.1%, 22.3%, and 20.8%. Majority of all profiles have been used in deriving the results in this study. ".*

Further, radiative calculations are performed once per hour. Why not compute all profiles? One of the main advantages of ground validation data is the high temporal resolution and a once per hour sampling lowers the accuracy of the data. If a subset of clouds is being examined, clouds could be missed and random errors could become large. It would be helpful to demonstrate that increased sampling does not impact the results (i.e. 6 times per hour or 10 times per hour).

*This is a good suggestion. The reason I used the hourly samples for the calculation is that the calculation is very time consuming. It will take a little over 2 months to calculate the radiative flux using the observations per minute, and a week for using samples per 10 minutes (6 times per hour). Also, the cloud properties usually do not change significantly within a few minutes, at least in the retrievals. Considering I have to repeat these calculations a few times for using data with all clouds, data with clouds from cloudsat+calipso/cloudsat/calipso, and some sensitivity studies, I decided to use the hourly data.*

*In response to reviewer's comment, I rerun all the calculations using samples 4 times per hour (every 15 minutes). I found the differences between using hourly data and using per 15 minutes data are small. The table below show their differences in monthly mean values at surface, with the maximum value at 0.6 Wm-2 (please see the table below). The differences at TOA are even smaller. All the findings in the submitted manuscript based on hourly data hold for those based on per 15 minutes data in the revision.*

*In the revision, all figures, tables, and text have been updated based on the calculations using per 15 minutes data. I also added a short clarification on the impact of temporal samples on the results, as "In this study, the radiation fluxes are computed and shown using profiles with 15-minute intervals (four out of 60 profiles), and there are 96 cases in a day except as otherwise stated. Daily means are computed based on the 96 values, and the monthly means are calculated based on the daily means. Computations are also made using profiles with 1-hour intervals (1 out of 60 profiles) to produce the daily means and monthly means. All the conclusions are the same, with the maximum differences in the CRF at the surface less than 0.6 Wm² and even less for those at the TOA."*

*The table shown here was included in the Appendix.*

- Table : Monthly mean cloud radiative forcing (CRF) at the surface for longwave (LW), shortwave (SW), and the combined LW and SW (all) with the clouds from the surface observations collected during the Surface Heat Budget of the Arctic Ocean (SHEBA) experiment and the differences between the CRF with clouds in the surface observations only identified from combined CloudSat and CALIPSO, CALIPSO, or CloudSat and the CRF from the clouds from the surface observations.

| | All clouds from surface observations with hourly data | | | (CloudSat+calipso)-clouds from surface with hourly data | | | (CloudSat+calipso)-clouds from surface with per 15 minutes data | | | All clouds from surface observations with per 15 minutes data | | |
|---|---|---|---|---|---|---|---|---|---|---|---|---|
| | LW | SW | all | LW | SW | all | LW | SW | all | LW | SW | all |
| Oct | 32.7 | -0.1 | 32.6 | -1.0 | 0.0 | -1.0 | -1.0 | 0.0 | -1.0 | 32.1 | -0.1 | 32.0 |
| Nov | 34.2 | 0.0 | 34.2 | -0.2 | 0.0 | -0.2 | -0.1 | 0.0 | -0.1 | 34.0 | 0.0 | 34.0 |
| Dec | 21.0 | 0.0 | 21.0 | 0.2 | 0.0 | 0.2 | 0.1 | 0.0 | 0.1 | 20.5 | 0.0 | 20.5 |
| Jan | 22.0 | 0.0 | 22.0 | 0.3 | 0.0 | 0.3 | 0.3 | 0.0 | 0.3 | 21.8 | 0.0 | 21.8 |
| Feb | 20.5 | -0.2 | 20.3 | 0.2 | 0.0 | 0.2 | 0.4 | 0.0 | 0.4 | 20.2 | -0.2 | 19.9 |
| Mar | 34.6 | -4.2 | 30.4 | 0.1 | 0.3 | 0.4 | -0.1 | 0.2 | 0.2 | 34.8 | -4.3 | 30.5 |
| Aprl | 42.5 | -12.6 | 29.9 | -0.9 | 0.6 | -0.3 | -0.8 | 0.6 | -0.2 | 42.7 | -12.7 | 30.0 |
| May | 43.3 | -22.3 | 21.1 | -3.0 | 2.9 | -0.1 | -3.1 | 2.9 | -0.2 | 43.9 | -22.6 | 21.4 |
| Jun | 44.4 | -34.5 | 10.0 | -2.5 | 3.6 | 1.1 | -2.6 | 3.2 | 0.6 | 45.0 | -34.5 | 10.5 |
| Jul | 43.9 | -61.0 | -17.1 | -3.4 | 6.1 | 2.7 | -3.0 | 5.5 | 2.4 | 44.0 | -61.0 | -16.9 |
| Aug | 59.8 | -33.9 | 25.9 | -2.8 | 4.0 | 1.2 | -3.4 | 3.9 | 0.5 | 59.8 | -33.4 | 26.4 |
| Sept | 63.2 | -8.2 | 55.0 | -3.0 | 0.4 | -2.6 | -2.9 | 0.4 | -2.5 | 63.0 | -8.2 | 54.9 |

2.

As stated in the uncertainty section, the results presented are based on a single-shot cloud mask (on profile at a time). The combined CloudSat and CALIPSO cloud masking products as well as the cited 2b-FLXHR-lidar products use averaged data along the satellite track to better detect clouds that might be missing in a one-shot case. The cloud mask results derived in the current study would represent a worst-case scenario for cloud detection errors from the satellite perspective. As demonstrated in figures 20 and 21, a large portion of clouds could be detected if a small change is made in the retrieval. This uncertainty, however, is not translated back to the radiation. It is, therefore, difficult to relate how this uncertainty impacts results and makes it hard for the reader to interpret. It would be helpful to take a month of data, such as when errors are large, to demonstrate a range of CRF uncertainty due to changes in cloud mask thresholds.

*Good suggestion. I thought to include this in the submitted manuscript, but did not because the manuscript was already too long and had too many figures/tables.*

*In the revision, I added the following tables in the supplement of the manuscript to show the uncertainty in the CRFs in monthly means for all months due to threshold changes in*

the CloudSat and Calipso cloud detection. I also included text in the revised manuscript on this, as

 "The impacts of the CALIPSO threshold changes on the monthly CRFs at the surface and the TOA are also investigated. Results show with increasing CALIPSO cloud detection capability, e.g. increasing thresholds, the LW (SW, total) monthly CRFs differences become smaller negative (positive, overall) (Table S#). Lidar with stronger cloud detection capability are desirable for better radiative flux estimations at the surface and the TOA. ".

 "The impacts of the CloudSat threshold changes on the monthly CRFs at the surface and the TOA show that the LW (SW, all) monthly CRFs differences are smaller negative (positive, overall) (Table S#). Radar more sensitive to the clouds near surface would help to produce more accurate radiative flux at the surface and the TOA".

"Combination of stronger lidar and more sensitive radar in the cloud detections produces smaller negative (positive, overall) bias for LW (SW, all) monthly CRFs differences (Table #)".

- Table : Monthly mean cloud radiative forcing (CRF) at the top of atmosphere for longwave (LW), shortwave (SW), and the combined LW and SW (all) with the clouds from the surface observations collected during the Surface Heat Budget of the Arctic Ocean (SHEBA) experiment and the differences between the CRF with clouds in the surface observations only identified from the CALIPSO and the CRF from the clouds from the surface observations. The CALIPSO cloud detection thresholds are 4, 5, and 6.

|  | All clouds from surface observations | | | CALIPSO-clouds from surface, with cloud detection threshold of 4 | | | CALIPSO-clouds from surface, with cloud detection threshold of 5 | | | CALIPSO-clouds from surface, with cloud detection threshold of 6 | | |
|---|---|---|---|---|---|---|---|---|---|---|---|---|
|  | LW | SW | all | LW | SW | all | LW | SW | all | LW | SW | all |
| Oct | 32.7 | -0.1 | 32.6 | -4.8 | 0.0 | -4.8 | -2.1 | 0.0 | -2.1 | -1.3 | 0.0 | -1.3 |
| Nov | 34.2 | 0.0 | 34.2 | -3.7 | 0.0 | -3.7 | -2.4 | 0.0 | -2.4 | -1.4 | 0.0 | -1.4 |
| Dec | 21.0 | 0.0 | 21.0 | -0.4 | 0.0 | -0.4 | -0.2 | 0.0 | -0.2 | 0.0 | 0.0 | 0.0 |
| Jan | 22.0 | 0.0 | 22.0 | -0.3 | 0.0 | -0.3 | 0.1 | 0.0 | 0.1 | 0.3 | 0.0 | 0.3 |
| Feb | 20.5 | -0.2 | 20.3 | -0.9 | 0.0 | -0.8 | -0.4 | 0.0 | -0.4 | -0.2 | 0.0 | -0.2 |
| Mar | 34.6 | -4.2 | 30.4 | -3.8 | 0.7 | -3.1 | -2.4 | 0.5 | -1.8 | -1.4 | 0.5 | -1.0 |
| Apr | 42.5 | -12.6 | 29.9 | -4.4 | 2.0 | -2.4 | -3.1 | 1.6 | -1.5 | -2.4 | 1.3 | -1.1 |
| May | 43.3 | -22.3 | 21.1 | -7.0 | 5.7 | -1.3 | -5.0 | 4.4 | -0.6 | -3.6 | 3.4 | -0.2 |
| Jun | 44.4 | -34.5 | 10.0 | -12.3 | 12.4 | 0.2 | -9.1 | 10.0 | 0.9 | -6.9 | 8.0 | 1.2 |
| Jul | 43.9 | -61.0 | -17.1 | -12.4 | 21.6 | 9.2 | -9.1 | 17.6 | 8.6 | -6.7 | 14.7 | 8.0 |
| Aug | 59.8 | -33.9 | 25.9 | -14.4 | 11.7 | -2.7 | -10.0 | 9.4 | -0.7 | -7.6 | 7.9 | 0.3 |
| Sept | 63.2 | -8.2 | 55.0 | -19.3 | 3.1 | -16.2 | -15.0 | 2.6 | -12.4 | -11.3 | 2.1 | -9.2 |

- Table : Monthly mean cloud radiative forcing (CRF) at the surface for longwave (LW), shortwave (SW), and the combined LW and SW (all) with the clouds from the surface

*observations collected during the Surface Heat Budget of the Arctic Ocean (SHEBA) experiment and the differences between the CRF with clouds in the surface observations only identified from combined CloudSat and the CRF from the clouds from the surface observations. The CloudSat's thresholds are threshold -10, threshold -15, and threshold-20.*

| | All clouds from surface observations | | | CloudSat-clouds from surface, with threshold - 10 | | | CloudSat-clouds from surface, with threshold - 15 | | | CloudSat-clouds from surface, with threshold - 20 | | |
|---|---|---|---|---|---|---|---|---|---|---|---|---|
| | LW | SW | all | LW | SW | all | LW | SW | all | LW | SW | all |
| Oct | 32.7 | -0.1 | 32.6 | -14.6 | 0.1 | -14.6 | -14.2 | 0.1 | -14.1 | -2.1 | 0.0 | -2.1 |
| Nov | 34.2 | 0.0 | 34.2 | -9.5 | 0.0 | -9.5 | -9.0 | 0.0 | -9.0 | -2.4 | 0.0 | -2.4 |
| Dec | 21.0 | 0.0 | 21.0 | -2.9 | 0.0 | -2.9 | -2.7 | 0.0 | -2.7 | -0.2 | 0.0 | -0.2 |
| Jan | 22.0 | 0.0 | 22.0 | -9.3 | 0.0 | -9.3 | -9.1 | 0.0 | -9.1 | 0.1 | 0.0 | 0.1 |
| Feb | 20.5 | -0.2 | 20.3 | -3.2 | 0.0 | -3.1 | -3.1 | 0.0 | -3.1 | -0.4 | 0.0 | -0.4 |
| Mar | 34.6 | -4.2 | 30.4 | -8.3 | 1.2 | -7.1 | -8.0 | 1.2 | -6.8 | -2.4 | 0.5 | -1.8 |
| Aprl | 42.5 | -12.6 | 29.9 | -16.1 | 4.9 | -11.1 | -15.9 | 4.9 | -11.0 | -3.1 | 1.6 | -1.5 |
| May | 43.3 | -22.3 | 21.1 | -23.5 | 12.0 | -11.5 | -22.6 | 11.5 | -11.1 | -5.0 | 4.4 | -0.6 |
| Jun | 44.4 | -34.5 | 10.0 | -16.8 | 11.5 | -5.4 | -16.4 | 11.1 | -5.3 | -9.1 | 10.0 | 0.9 |
| Jul | 43.9 | -61.0 | -17.1 | -15.9 | 19.2 | 3.3 | -15.7 | 19.0 | 3.3 | -9.1 | 17.6 | 8.6 |
| Aug | 59.8 | -33.9 | 25.9 | -22.0 | 13.7 | -8.4 | -20.9 | 13.0 | -7.9 | -10.0 | 9.4 | -0.7 |
| Sept | 63.2 | -8.2 | 55.0 | -16.2 | 2.0 | -14.2 | -15.3 | 1..9 | -13.5 | -15.0 | 2.6 | -12.4 |

- *Table : Monthly mean cloud radiative forcing (CRF) at the surface for longwave (LW), shortwave (SW), and the combined LW and SW (all) with the clouds from the surface observations collected during the Surface Heat Budget of the Arctic Ocean (SHEBA) experiment and the differences between the CRF with clouds in the surface observations only identified from combined CloudSat and CALIPSO with different thresholds and the CRF from the clouds from the surface observations.*

| | All clouds from surface observations | | | (CloudSat+CALIPSO)-clouds from surface, CloudSat threshold of standard-15, and CALIPSO threshold of 5 | | | (CloudSat+CALIPSO)-clouds from surface, CloudSat threshold of standard-20, and CALIPSO threshold of 6 (maximum detection) | | | (CloudSat+CALIPSO)-clouds from surface, CloudSat threshold of standard-10, and CALIPSO threshold of 4 (minimum detection) | | |
|---|---|---|---|---|---|---|---|---|---|---|---|---|
| | LW | SW | all | LW | SW | all | LW | SW | all | LW | SW | all |
| Oct | 32.7 | -0.1 | 32.6 | -1.0 | 0.0 | -1.0 | -0.6 | 0.0 | -0.6 | -1.8 | 0.0 | -1.8 |
| Nov | 34.2 | 0.0 | 34.2 | -0.2 | 0.0 | -0.2 | -0.1 | 0.0 | -0.1 | -0.7 | 0.0 | -0.7 |

| | | | | | | | | | | | | |
|---|---|---|---|---|---|---|---|---|---|---|---|---|
| Dec | 21.0 | 0.0 | 21.0 | 0.2 | 0.0 | 0.2 | 0.2 | 0.0 | 0.2 | 0.2 | 0.0 | 0.2 |
| Jan | 22.0 | 0.0 | 22.0 | 0.3 | 0.0 | 0.3 | 0.3 | 0.0 | 0.3 | 0.1 | 0.0 | 0.1 |
| Feb | 20.5 | -0.2 | 20.3 | 0.2 | 0.0 | 0.2 | 0.1 | 0.0 | 0.1 | 0.3 | 0.0 | 0.4 |
| Mar | 34.6 | -4.2 | 30.4 | 0.1 | 0.3 | 0.4 | 0.5 | 0.3 | 0.7 | -0.2 | 0.3 | 0.1 |
| April | 42.5 | -12.6 | 29.9 | -0.9 | 0.6 | -0.3 | -0.7 | 0.4 | -0.2 | -1.4 | 0.8 | -0.6 |
| May | 43.3 | -22.3 | 21.1 | -3.0 | 2.9 | -0.1 | -2.0 | 2.1 | 0.1 | -4.3 | 3.8 | -0.5 |
| Jun | 44.4 | -34.5 | 10.0 | -2.5 | 3.6 | 1.1 | -1.8 | 3.0 | 1.2 | -4.1 | 4.7 | 0.7 |
| Jul | 43.9 | -61.0 | -17.1 | -3.4 | 6.1 | 2.7 | -2.4 | 4.9 | 2.5 | -4.8 | 7.6 | 2.8 |
| Aug | 59.8 | -33.9 | 25.9 | -2.8 | 4.0 | 1.2 | -1.7 | 3.3 | 1.6 | -4.8 | 5.2 | 0.4 |
| Sept | 63.2 | -8.2 | 55.0 | -3.0 | 0.4 | -2.6 | -2.3 | 0.3 | -2.0 | -3.8 | 0.5 | -3.3 |

3.

Something seems off about the optical depth calculations shown in Figure 1. Previous studies using SHEBA data display column optical depth ranges up to ~30, with most clouds having column optical depths < 10 (Turner 2005; Zuidema et al 2005). If the vertical resolution is 63 m., Figure 1 shows optical depths greater than 5 for each vertical bin in the 15-20 bins below cloud top. This would lead to a column optical depth > 100 and does not seem physical for such a geometrically thin cloud. Given that the optical depth is used in the radiative transfer calculations as well as the CALIPSO cloud mask it would also lead to large uncertainty in the results as it prevents the CALIPSO cloud mask detections. This needs to be investigated to make sure the correct calculations have been made throughout the study.

Turner, D. D. (2005). Arctic Mixed-Phase Cloud Properties from AERI Lidar Observations: Algorithm and Results from SHEBA, Journal of Applied Meteorology, 44(4), 427-444

Zuidema, P., Baker, B., Han, Y., Intrieri, J., Key, J., Lawson, P., Matrosov, S., Shupe, M., Stone, R., & Uttal, T. (2005). An Arctic Springtime Mixed-Phase Cloudy Boundary Layer Observed during SHEBA, Journal of the Atmospheric Sciences, 62(1), 160-176

*In Figure 1 of the submitted manuscript, Figure 1b is the accumulated cloud optical thickness from the top of atmosphere, not the vertical distribution. The caption "(b) integrated optical thickness from top of atmosphere for CALIPSO" is confusing.*

*In the revision, I added a new figure to show the CALIPSO cloud optical thickness vertical distribution, and also modified the figure caption. I also added the two references in the revision.*

*The new figure is shown below.*

[Figure]

Figure 1: Simulated (a) CloudSat reflectivity and (b) CALIPSO cloud optical thickness, and 3) accumulated cloud optical thickness from top of atmosphere for CALIPSO on November 21, 1997 during the Surface Heat Budget of the Arctic Ocean (SHEBA) experiment.

5.

There are a large number of tables and figures in the paper. Figures or tables that are not necessary could be removed or added to supplemental. For example, the surface albedo

figures are not needed as their source is described in the text.  Further, the tables showing the raw numbers for vertical profiles could be included in the supplemental.

*Thanks for this good suggestion. In the revision, I moved a few figures and most of the tables to the supplemental section.*

Specific Comments

Paper Title: The title is too vague and suggests a global study when the spatial sampling is limited to ground validation sites in the Arctic. I would include the Arctic in the title to make it more specific or mention the use of SHEBA data.

*Thanks for the suggestion. The paper title has been changed to "Impacts of Active Satellite Sensors' Low-level Cloud Detection Limitations on Cloud Radiative Forcing in the Arctic".*

Pg 1 Line 22: Change "modulator of the radiation flux" to "modulator of radiation".

*Changed.*

Pg 4 Line3:  I would make this clearer that phrases such as "with every 63 m from 150 to 1050 m" is talking about changes in vertical resolution.

*Changed to "and the vertical coverage is from 150 m to 22950 m with vertical resolution of 63 m from 150 to 1050 m and vertical resolution of 100 m above 1050 m."*

Fig 1b: A log-scale color bar would make some of the finer details of the cloud optical depth pop out a little more and limit the saturated COD above 5 (see comment 3 above as well).

*I kept the color bar. I added a new figure to show the COD vertical distribution. Please see the response to comment 3.*

Pg 7 Line 5. I would move the description of the radiative terms being calculated after the description the two sets of profile experiments.

*Done.*

Pg 7: Line 18: The use of "cloud layers" or "layers" gets jumbled here. I would describe cloud layers as layers with cloud data and total layers (50 or 125) as just "layers".

*Changed to "Streamer can simulate the radiation fluxes for up to 50 layers with cloud data. In this study, only layers from 150 m to 12.0 km are simulated. There are 125 layers from 150 m to 12.0 km in the retrieved cloud data sets, so that there are potential 125 layers with*

*cloud data at maximum. Every layer with cloud data below 2.0 km was included in the Streamer input file…"*

Pg 9: Line 21: Please introduce Figure 6 first.

*Added "Figure 6 shows the cloud vertical distribution from CALIPSO, CloudSat, and combined CALIPSO and CloudSat and their differences with surface observations."*

Pg 9 Line 28: remove "very" from high optical depths.

*Done.*

Pg 11 Line 2: Should this reference Fig 6e?

*You are correct. Correction has been made in the revision.*

Fig 8: Is "all" the surface observations? This should be made clear.

*"All" has been replaced with "Surface observations" in the figure. The figure is updated.*

Pg 12 Line 12. With a difference of only a few percent, I would mention that the combined retrieval captures the majority of clouds above 1 km.

*The sentenced has been changed to "As the result, the combined CloudSat and CALIPSO detects most of the clouds that surface observation sees above 2 km, and majority of the clouds above 1 km."*

Fig 6. The Julian Day labeling is a bit confusing and would be easier to interpret if months could be added.

*Changes have been made based on Reviewer's suggestion. This figure has been moved to the supplement.*

Pg 16 Line 10: This is mostly due to the lack of sunlight during many months leading to the LW heating being the dominant term.

*Good point. Thanks. Added this sentence in the revision "This is mostly due to the lack of sunlight during winter months leading to the LW heating being the dominant term."*

Pg 28 Line 5: This caveat along with the one for CloudSat needs to be listed in the cloud mask description in section 2.

*Good suggestion. Added in the section 2 ". It should be noted that a more sophisticated detection scheme is used in the operational CALIPSO cloud mask product (Winker et al. 2009) and operational CloudSat cloud mask (Marchand et al. 2008), and results in this study should*

*not be treated as the results from operational products.", and kept this in section 3 to emphasize the point.*

---

## Author Comment (AC2)

Response to Reviewer's Comments:
(*Authors' responses are in italic and in blue color*)

Reviewer 2:

This study examined the potential biases in the joint CloudSat/CALIPSO cloud mask and radiative fluxes retrievals. Due to the difficulties in distinguishing the cold surface from clouds by passive sensors in the polar regions, the joint CloudSat/CALIPSO observations are critical to study polar clouds and their radiative effects. Although the limitations of the CloudSat CPR and the CALIPSO cloud observations and retrievals are well known, this is the first study to systematically quantify the potential biases in the CloudSat/CALIPSO cloud cover and radiative forcings due to such limitations.

This study uses the ground-based MMCR retrievals during SHEBA as input to the QuickBeam radar simulator to simulate cloud profiles from the perspective of CloudSat CPR and the CALIPSO CALIOP. The effects of ground clutter of the CPR signals are carefully approximated. Adding the approximate ground clutter together with the simulated CALIOP integrated signal attenuation, the author identifies the missing clouds in the simulated profiles using a simplified version of the joint CloudSat and CALIPSO cloud mask. Indepth and detailed analysis of the contribution to the missing clouds and the cloud radiative forcing (CRF) from clouds of different phase are presented. The author finds monthly CRF uncertainties up to 2.7 W/m2 at the surface and 4.0 W/m2 at TOA and the uncertainties are up to 30 W/m2 in individual cases.

The findings of this research provide a more solid basis for the studies of the Arctic cloud and cloud radiative effects. This manuscript is well suited for publication at the Atmospheric Chemistry and Physics. At this point, it still requires careful revision before publication, especially for Section 3.2, mostly with clarification and improvements to presentation. Detailed comments are listed below.

*We thank the reviewer for the careful review, positive comments, and constructive feedbacks. Revision based on the reviewer's comments has greatly improved the manuscript.*

General comments:
Maybe consider moving redundant information to the supplement materials, especially the long tables and Figures from previous publications. I do appreciate the author having all the data in the table for easy and quick reference but having them in the manuscript sometimes disrupts the flow of the presentation.

*Thanks for this good suggestion. In the revision, I moved a few figures and most tables to the supplemental section.*

Specific comments:
P2, Line 23: "CALIPSO" is capitalized here. Please use consistent abbreviations throughout the manuscript.

*Checked and found a few in lower case, and changed them to upper case in the revision.*

P2, Line 23: "high-level clouds", this sentence is ambiguous because the signal attenuation issues can happen for clouds at any level.

*I changed it to "the clouds above the low-level clouds". Hopefully, this eliminates the confusion.*

P3, Line 30: It should be Shupe et al. 2006.

*Corrected.*

P4, Line 6-7: Is there an estimate of how many profiles are excluded? Maybe adding a qualitative estimate of how such selection will affect the total CRF in later sections as well? Including such assessments in the manuscript will help the readers to interpret the results of this study more accurately.

*The focus of study is the impact on the cloud detection and cloud radiative forcing. It is understandable to exclude vertical profiles including snow, drizzle, liquid cloud+drizzle, rain, haze, or uncertain retrievals. There might also be larger uncertainties associated with these retrievals, and the radiative transfer models also have higher uncertainties to simulate the reflectivity and radiative fluxes for these cases. For this reason, this study will not be able to provide a quantitative estimate of the uncertainty. This might be a good topic for future work.*

*Meanwhile, qualitative estimate can be made. In the revision, I provided the percentages of these profiles to all profiles for each month to show how many cases have been excluded. Also, in the discussion section, I emphasized this limitation and its possible impact on the total CRF "The study focuses on the impacts of active satellite sensors' low-level cloud detection limitations on cloud radiative forcing, so vertical profiles including snow, drizzle, liquid cloud+drizzle, rain, haze, or uncertain retrievals were excluded in calculating the CRFs. There are over 30,000 profiles in every month from October 1997 to September 1998, except that the total profile numbers are around 15,800 in October, which includes October 1997 and October 1998. Of all the profiles in every month from October 1997 to September 1998, these profiles with snow, drizzle, liquid cloud+drizzle, rain, haze, or uncertain retrievals account for 11.6%, 17.0%, 7.3%, 9.0%, 7.3%, 10.5%, 9.1%, 4.9%, 10.1%, 22.3%, and 20.8%. Majority of the profiles have been used in deriving the results in this section. ".*

P4, Line 16: It is mentioned on P7 that solid hexagonal column is used for ice particles in radiative transfer code. Is that consistent with the Quickbeam setting here? By default, Quickbeam uses ice spheres. If solid column is used here, are other parameters adjusted as well, such as for area/mass ratio?

*You are correct that ice crystals in Quickbeam are modeled as "soft spheres" meaning the diameter of a given sphere is the same as the maximum dimension of the corresponding ice crystal. In the Quickbean simulations, I used the effective radius as the radius of the ice sphere. In the Streamer simulation, to better describe the ice particle habit, I used the solid hexagonal*

*column, with the same effective radius as that in Quickbeam. With this setting, I expect the simulations in Quickbeam and Streamer are consistent. I clarified this in the revised manuscript.*

*"The mixing ratios were set to use the cloud ice/liquid water content, and the cloud phase and cloud effective radius come from the 1-min-interpolated cloud phase and cloud effective radius." "Though the ice cloud particle shape is different from that in QuickBeam, the ice cloud effective radius is the same as what used in the QuickBeam simulation."*

*Furthermore, I did some tests on the Quickbeam simulations with effective radius changing from effective radius \* 0.8 to 1.2\*effective radius, the derived CloudSat ice cloud cover at most layers is within 1.5% of the original cloud amount. The impacts on the surface radiative fluxes are negligible due to small optical thickness of the ice clouds.*

P4, Line 18: Haynes et al. 2007 for the BAMS paper about Quickbeam.

*Added "Haynes et al. 2007". I referred two papers by Haynes in this study both in 2007.*

P5, Line 12-13: Does this statement refer to subtracting the estimated mean clutter from the received power of the lowest four bins? Marchand et al. (2008) mentioned it as preliminary. Was this approach shown to be effective in later studies or used in later version of the CloudSat cloud mask processing?

*My understanding is a return would be considered as coming from a hydrometeor if it is higher than the mean radar-measured noise power near the surface, here defined as the 99th percentile of the clear-sky returns as shown in Marchand et al. (2008). With the lower mean radar-measured noise power near the surface, CloudSat supposes to detect more cloud near the surface. My understanding is that this approach is used in CloudSat Hydrometeor Mask version4.*

*You are correct that the adjustment we made in this study only appears in Marchand et al. (2008). I cannot find any peer-reviewed publications that confirm whether this adjustment is effective or not. Attempt to get information from the algorithm developer through personal communication has also failed. Only thing I can find is a document on the CloudSat website at* [https://www.cloudsat.cira.colostate.edu/cloudsat-static/info/dl/2b-geoprof/2B-GEOPROF_PDICD.P1_R05.rev0__0.pdf](https://www.cloudsat.cira.colostate.edu/cloudsat-static/info/dl/2b-geoprof/2B-GEOPROF_PDICD.P1_R05.rev0__0.pdf)*, e.g. on page 12. Again, there is no clear information whether or how this update has been implemented in the CloudSat R05.*

*In this study, sensitivity study has been carried out to test how the CloudSat cloud detection varies with different thresholds used in the lowest 5 CloudSat range bins. Based on the results, the -15 dBZe adjustments made in the lower altitude are expected to lead to higher cloud amount detected from the CloudSat. Without the -15 dBZe adjustment, the CloudSat would detect even less cloud in the lower altitude compared to the surface observations, thus leading to even higher radiative flux bias.*

*To clarify, the text has been changed to "A layer was flagged as cloud if the simulated CloudSat reflectivity at a layer is larger than the mean radar-measured noise power at that layer, which is the 99th percentile of the clear-sky returns as in Figure 7 of Marchand et al. (2008), Figure 2 in this paper. The vertical line of the 99th percentile of the clear-sky returns serves as thresholds to detect cloud by CloudSat at all layers. This threshold provides a very stringent requirement for cloud detection, especially for cloud detection near the surface. Since it was suggested that the CloudSat cloud detection capability would be improved with lower mean radar-measured noise power near the surface (Marchand et al. 2008), in this study, the thresholds was further lowered by 15 dBZe in the lowest 5 range bins (lower than 960 m) than the 99th percentile of the clear-sky returns while higher than or equal to -26 dBZe (Marchand et al. 2008)."*

*In section 3.3 of the revised manuscript, I added "In this study, the CloudSat cloud detection threshold was further lowered by 15 dBZe in the lowest 5 range bins (lower than 960 m) than the 99th percentile of the clear-sky returns while higher than or equal to -26 dBZe as indicated in Marchand et al. (2008). Without this adjustment, the CloudSat would detect less cloud between 550 m and 960 m, which might lead to even larger radiative flux bias."*

P8, Line 8-11: These two sets are referred to frequently later in the manuscript. Maybe it is useful to have an abbreviation/code name for each of them? The efforts to describe them accurately make sentences long, convoluted, and confusing. Similarly, it might be worthwhile to find a more succinct way to address cases when clouds of a specific phase are missed by the combined CloudSat/CALIPSO mask.

*I do not see P8 Line 8-11 in PDF version of the submitted manuscript. I think it is P7, Line 8-11. In the revision, the combined CloudSat/CALIPSO is abbreviated as CC. Surface cloud profile is used for the complete cloud profiles from the surface observations, including cloud classification, cloud effective radius, and cloud water content. Satellite cloud profile, e.g. CALIPSO/CloudSat/CC cloud profile, is used for the subset of the cloud profiles including only those layers identified as clouds in the space-based radar, lidar, or combined lidar and radar cloud mask.*

P9, Line 8-11: Maybe specify the sign of the difference here as well? I can tell from the captions of Fig. 10 and 16 but it will be easier for the readers to follow to clarify.

*Added "Positive CRF at surface (TOA) indicates that the clouds warm the surface (earth-atmosphere system) relative to the clear skies, and negative CRF indicates that clouds cool the surface (earth-atmosphere system).", and also "In this paper hereinafter, the differences in the CRFs means the differences between CRFs from satellite cloud profiles and the CRFs from surface cloud profiles. Positive differences indicate more warming effect, and negative differences indicate less warming effect or more cooling effect".*

Fig. 6 and 9: Maybe use thicker lines?

*Changed has been made in Figure 8 (not 6) and 9.*

P16, Line 10: The time period of negative CRF from SHEBA is much shorter than the results by Kay and L'Ecuyer (2013). The SHEBA CRF is likely biased because most of its albedo observations were taken mostly locally from ice. It is true that they used a 200 m line to sample more surface types, but it was not likely going to represent the extent of open water over a larger area. The length of negative CRF may vary with latitude and the time span of surface melt/open water. So, the SHEBA CRF does show the typical seasonal cycle of the CRF in the Arctic Ocean but might not be an accurate representation of the Arctic Ocean in summer.

*Agreed. Also during the SHEBA year, the surface albedo may be lower than the albedo in recent years. Discussion was added in the section , as "The SHEBA data have been invaluable in studying the Arctic climate system. They were collected in a limited area at a certain time in a year, thus they may have limitations in representing the whole Arctic Ocean in a longer time scale. This is indicated by the shorter time period of negative CRF shown in this study than the results by Kay and L'Ecuyer (2013). Among all possible causes, the surface albedo data during SHEBA may represent the sea ice more than larger area with open water. Results of this study therefor are subject to uncertainties due to the spatial and temporal limitations of the data."*

Fig. 10 and 16: Maybe change to two panels, one for CRF for all profiles, and one for the difference of CC from all? The differences are difficult to tell in a single plot.

*Good suggestion. I have updated the figures into two panels.*

P18, Line 6: How about adding the last sentence of caption in Fig. 11 here too? It helps to remind the reader that CRF differences within 2 W/m2 are not included in the following analysis.

*It has been added, "Please note that cases with absolute differences less than 2 Wm-2 are the majority while excluded in the histogram in Figure 11."*

P18, line 8-9: This sentence is confusing, please rephrase.

*It has been rephrased to "the same conclusion holds for clouds being ice, liquid or mixed phase clouds".*

P19, Line 14: Does this mean the low-level clouds around 1 km is still opaque enough that the effective emitting temperature are close to when the clouds below 1 km are included?

*Yes, you are correct. I think this is the cause for those cases with very small LW CRFs differences. I revised this part as "In the winter months, temperature inversions are relatively weaker under cloudy conditions (Figure 14a), the cloud effective temperature is close to the surface temperature in a well-mixed boundary layer (Tjernström and Graversen 200). Removal of some of these lower-level clouds would not likely greatly change the downward longwave radiation at*

*the surface as long as there are clouds with similar effective temperature on top of these undetected clouds, which leads to the small longwave CRF differences in the winter months."*
*I also updated Figure 14a.*

P19, Line 15: Figure 11, S6?

*Corrected.*

P19, Line 15-17: this sentence is long and confusing. Please rephrase.

*The sentence is rephrased as "The longwave CRF differences at the surface are close to zero in the winter months, and tend more towards large negative values in the summer months especially with liquid and mixed phase clouds near the surface (Figure 11, S6). In the summer months, the longwave CRF differences become larger negative with the increasing cloud optical thickness below 1 km and those clouds not being detected by the combined CloudSat and CALIPSO (Figure 13)."*

Fig. 13 15, and 18: These scatter plots are pretty noisy. Because the optical depth is related to the fraction of reduction of incoming radiative fluxes and not with the absolute values of the fluxes, normalizing the CRF differences with the solar zenith angle for SW and Ts^4 for LW may produce a cleaner fit.

*Thanks for the suggestion. I tried the SW CRF with normalization with solar zenith angle. It does not improve the figure much. Also the relations/equations of CRF differences and cloud optical thickness may provide some useful information for the modelers for parameterization to account for the impact of sensor detection limitation in the current form. Based on these considerations, I kept these figures in the current form.*

P20, Line 1: I think the temperature of clouds below 1 km should be compared to the effective emitting temperature of the clouds above 1 km instead of to the surface temperature. The LW CRF being small in winter suggests the clouds above 1 km emit at a temperature cloud to the temperature within 1 km. Fig. S1 shows a liquid layer above but close to 1 km, which is consistent with LW CRF being small. The question is how representative this case is in winter.

*I think you are right. The argument in the old manuscript is not valid. This Please see my response to your comment regarding P19 Ln 14. Looking at figure 3b in Tjernström and Graversen 2009, I think this is quite common during the SHEBA year.*
*Tjernström, M. and Graversen, R. G.: The vertical structure of the lower Arctic troposphere analysed from observations and the ERA-40 reanalysis, Q. J. R. Meteorol. Soc., 135(639), 431–443, doi:10.1002/qj.380, 2009.*

P22, Line 9-10: This statement is not exactly accurate. The clouds are brighter than the surface in the summer mainly because the surface has more open water, more ponds, and less snow-covered ice rather than clouds being brighter.

*Agreed. The sentence has been changed to "The clouds in the summer months reflect more shortwave radiation back to space than surface especially in July when the surface albedo is low due to melt ponds and more open water, such that the shortwave CRFs at TOA are all negative in the summer months."*

P22, Line 12-13: Is it possible that the LW effective emitting temperature at TOA is much higher in the atmosphere that the BL inversion is not relevant at tall?

*This is possible, especially in the summer months when water vapor content is higher. The argument in the manuscript was mainly to explain the positive CRFs in the winter months. I have changed the sentence to "This is possibly due to colder effective cloud top temperature than the effective emitting temperature from the combined atmosphere and surface, even in the winter months when surface temperature inversions are common."*

P23, Line 9-10: Maybe this is another sign that the LW emitting temperature is above 1 km?

*You are correct. I agree that most of the LW emitting temperature is above 1 km in summer months considering the very small differences, and the omission of the lower clouds might be negligible. If I can point out one thing, that will be the differences are very small but negative in all summer months. This might indicate the omission of the lower clouds may have very small impact. In the wintertime, the omission of the lower level clouds has little impact possibly due to possibly isothermal temperature in the boundary layer. Anyway, I added in the revision "The differences in the longwave CRFs are near zero in all months, with the maximum magnitude in July at $-0.7$ $Wm^{-2}$, which indicate that the contribution of clouds not detected near the surface are insignificant to the LW radiation at TOA."*

P24, Line 7: "... determine the CRFs" to "... determine the CRF changes".
*Corrected.*

P25, Line 11-12: Add comma between "Cloud especially ..." and add comma between "… near the surface help brighten …".
*Done.*

P25, Line 12: Remove the second "larger".
*Done.*

P25, Line 13: Remove "more".
*Done.*

P25, Line 14: Remove "larger".

*Done.*

Fig. 20, 21: use thicker lines and larger fonts.
*Done.*

P26, Line 22: Is there a reason for Quickbeam to perform better in ice clouds? Comparing Fig. S1 and Fig 19, the large reflectivity differences are in a slightly shallower layer than the retrieved liquid layer in Fig. S1. Maybe the larger differences in reflectivity of liquid clouds are related to the uncertainties in the height of phase transition? Using the refractive index of water for ice clouds will increase reflectivity significantly. If you set the places with large liquid reflectivity differences to ice phase for the case in Fig. S1, would it get rid of the large reflectivity differences? If so, it might point to phase retrieval error.

*I tested the reviewer's suggestion. I set the places with large liquid reflectivity differences to ice phase for the case in Fig S1, it did not reduce the large reflectivity differences.*

*I further tested other possibilities. I changed the liquid cloud effective radius and did the simulation. With smaller liquid cloud effective radius, the differences became smaller.*

*This probably shows there is higher uncertainty in the liquid cloud effective radius retrivals for this specific case. I added a short discussion in the revised manuscript on this subject, as "Further tests show the large differences for liquid cloud became smaller when smaller liquid cloud effective radius was used in the simulation. This might indicate there are higher uncertainties in the liquid cloud effective radius retrievals for this specific case".*

---

## Author Response (AR2)

Response to Reviewer's Comments:
(*Authors' responses are in italic and in blue color*)

Reviewer 1:

I would like to thank the authors for a clear response to my questions. I think the new additions significantly improve the uncertainty discussion. Further, thank you for clarifying my misunderstanding of the optical depth. I think the revised figure that includes the integrated and raw values is useful. I only have a few minor clarifications that I would suggest before the final submission. Specifically, I am still a bit confused by response related to the omitted cloud types. My comments are below.

*I appreciate the reviewer's comments and suggestions. I totally agree with the reviewer that the manuscript has improved greatly following the reviewers' suggestions.*

Page 11 line 1: Fix "the cloudsat"

*It seems there is no error here. I double checked the "CloudSat" throughout the manuscript and made sure they are all correctly spelled.*

Line 15 pg 20 :

"Of all the profiles in every month from October 1997 to September 1998, the profiles with snow, drizzle, liquid cloud+drizzle, rain, haze, or uncertain retrievals account for 11.6%, 17.0%, 7.3%, 9.0%, 7.3%, 10.5%, 9.1%, 4.9%, 10.1%, 22.3%, and 20.8%. Majority of all profiles have been used in deriving the results in this study."

The percentages for these categories do not match. There are eleven numbers and only six categories mentioned. Further, the summation exceeds 100%. Please clarify this part and provide the proper respective categories. It would be helpful to include the percentage of the 10 categories and also add an all-inclusive statement, such as "XX% of all profiles were included in the CRF analysis and XX% were omitted due to uncertain retrievals". Finally, I would provide these percentages on Page 4 line 5.

*Sorry the confusion. Actually, these percentages are exactly the reviewer suggested. I just did not explain them clearly and missed one number.*

*These percentages are the ratios of numbers of profiles that include any of these 6 categories (snow, drizzle, liquid cloud+drizzle, rain, haze, or uncertain retrievals) to all profiles for each month from October 1997 (including October 1998 for October ratio) to September. There should be 12 numbers. However, I missed the last one (September 1998). In the revised manuscript, I added the number for September 1998 (12 numbers for 12 months, and October include October 1997 and October 1998), and modified the text to clarify.*

*The revised texts are: "The study focuses on the impacts of active satellite sensors' low-level cloud detection limitations on cloud radiative forcing, so vertical profiles including snow, drizzle, liquid cloud+drizzle, rain, haze, or uncertain retrievals were excluded in calculating the CRFs. There are over 30,000 profiles in every month from October 1997 to September 1998, except that the total profile numbers are around 15,800 in October, which includes October 1997 and October 1998. Of all the profiles in each month, the profiles including any of the 6 conditions, i.e. snow, drizzle, liquid cloud+drizzle, rain, haze, or uncertain retrievals, account for 11.6%, 17.0%, 7.3%, 9.0%, 7.3%, 10.5%, 9.1%, 4.9%, 10.1%, 22.3%, 20.8%, and 10.6% from October 1997 to September 1998. Majority of the profiles, equal to or more than 77.7% in all the months, have been used in deriving the results in this study."*

Page 17 Line 6. Add Fig 10 reference.

*Added.*

Fig 10 and 11: "Please note that cases with absolute differences less than 2 Wm-2 are the majority while excluded in the histogram in Figure 10".

Could you include a percentage as " the majority" is vague.

*Numbers have been added. In the last round of revised manuscript, the 2 Wm-2 has been updated to 1 Wm-2 in plotting the figure, but the text was not updated. In this revised manuscript, I updated this change, and also added the ratios of cases outside 1 Wm-2 to all cases.*

Section 3.3 "It should be noted that a more sophisticated detection scheme…."

You could combine the CloudSat and CALIPSO statements to reduce redundancy of this statement.

*The text has been cleaned up.*